# Fairness Under Demographic Scarce Regime

**Patrik Joslin Kenfack**  *patrik-joslin.kenfack.1@ens.etsmtl.ca*
*ÉTS Montréal, Mila*

**Samira Ebrahimi Kahou**  *Samira.Ebrahimi.Kahou@gmail.com*
*University of Calgary, Mila*
*Canada CIFAR AI Chair*

**Ulrich Aïvodji**  *Ulrich.Aivodji@etsmtl.ca*
*ÉTS Montréal, Mila*

**Reviewed on OpenReview:** *https: // openreview. net/ forum? id= TB18GOw6Ld*

## Abstract

Most existing works on fairness assume the model has full access to demographic information. However, there exist scenarios where demographic information is partially available because a record was not maintained throughout data collection or for privacy reasons. This setting is known as *demographic scarce regime*. Prior research has shown that training an attribute classifier to replace the missing sensitive attributes (*proxy*) can still improve fairness. However, using proxy-sensitive attributes worsens fairness-accuracy tradeoffs compared to true sensitive attributes. To address this limitation, we propose a framework to build attribute classifiers that achieve better fairness-accuracy tradeoffs. Our method introduces uncertainty awareness in the attribute classifier and enforces fairness on samples with demographic information inferred with the lowest uncertainty. We show empirically that enforcing fairness constraints on samples with uncertain sensitive attributes can negatively impact the fairness-accuracy tradeoff. Our experiments on five datasets showed that the proposed framework yields models with significantly better fairness-accuracy tradeoffs than classic attribute classifiers. Surprisingly, our framework can outperform models trained with fairness constraints on the true sensitive attributes in most benchmarks. We also show that these findings are consistent with other uncertainty measures such as conformal prediction. The source code is available at https://github.com/patrikken/fair-dsr.

## 1 Introduction

Mitigating machine learning bias against certain demographic groups becomes challenging when demographic information is wholly or partially missing. Demographic information can be missing for various reasons, e.g., due to legal restrictions, prohibiting the collection of sensitive information of individuals, or voluntary disclosure of such information. As people are more concerned about privacy, reluctant users will not provide sensitive information. As such, demographic information is available only for a few users. A *demographic scarce* regime was the term used by Awasthi et al. (2021) to describe this particular setting. The data in this setting can be divided into two sets: $\mathcal{D}_1$ and $\mathcal{D}_2$. The dataset $\mathcal{D}_1$ does not contain demographic information, while $\mathcal{D}_2$ contains both sensitive and non-sensitive information. The goal is to train a fair classifier with respect to different (unobserved) demographic groups in $\mathcal{D}_1$. Without demographic information in $\mathcal{D}_1$, it is more challenging to enforce group fairness notions such as *statistical parity* (Dwork et al., 2012) and *equalized odds* (Hardt et al., 2016). Algorithms to enforce these notions require access to sensitive attributes to quantify and mitigate the model's disparities across different groups (Hardt et al., 2016; Agarwal et al., 2018). However, having access to another dataset where sensitive attributes are available gives room to train a sensitive attribute classifier that can serve as a *proxy* for the missing ones. We are interested in understanding what level of fairness/accuracy one can achieve if proxy-sensitive attributes are used to replace

the true sensitive attributes and properties of the sensitive attribute classifier and the data distribution that influences the fairness-accuracy tradeoff.

In their study, Awasthi et al. (2021) demonstrated a counter-intuitive finding: when using proxy-sensitive attributes, neither the highest accuracy nor an equal error rate of the sensitive attribute classifier has an impact on the accuracy of the bias estimation. Although Gupta et al. (2018) showed that improving fairness for the *proxy* demographic group can improve fairness with respect to the true demographic group; it remains unclear how existing fairness mechanisms would perform in the presence of proxy-sensitive attributes and how the fairness-accuracy tradeoff is impacted. We show that existing fairness-enhancing methods (Hardt et al., 2016; Agarwal et al., 2018) can be robust to noise introduced in the sensitive attribute space by the proxy attribute classifier, i.e., unfairness can be mitigated when proxy attributes are used instead of the sensitive attribute. However, the fairness-accuracy tradeoff worsens when fairness constraints are imposed on these proxy-sensitive attributes. We aim to provide insights into the distribution of sensitive attributes that can yield better fairness and accuracy performances. We hypothesize that the uncertainty of the sensitive attribute classifier plays a critical role in improving fairness-accuracy tradeoffs on downstream tasks. Specifically, we argue that samples with *reliable* demographic information should be used to fit fairness constraints, backed up by the intuition that these samples are *easier* to discriminate against, while samples with uncertain demographic information are already hard to discriminate against. Validating this hypothesis requires effective uncertainty estimation in the sensitive attribute space.

In this paper, we propose a framework to handle **Fair**ness under **D**emographic **S**carce **R**egime (FairDSR). FairDSR consists of two phases. In the first phase, we construct an uncertainty-aware Deep Neural Network (DNN) model to predict demographic information. The training is semi-supervised using self-ensembling (Tarvainen & Valpola, 2017), and the uncertainty of the predictions is measured and improved during the training using Monte Carlo dropout (Gal & Ghahramani, 2016). The first phase outputs for each data point, the predicted sensitive attribute, and the uncertainty of the prediction. In the second phase, the classifier for the target task is trained with fairness constraints w.r.t to the predicted sensitive attributes. However, fairness constraints are imposed only on samples whose sensitive attribute values are predicted with low uncertainty. Our results rely heavily on dropout regularization and the effectiveness of the uncertainty measure. Existing uncertainty measures compute prediction uncertainty using the model in a certain epoch, e.g., in the last training epoch. However, the uncertainty measure can vary for different models in different training epochs due to different perturbations, such as network dropout, input noise, and data ordering. Hence, we adopt self-ensembling to compute more consistent uncertainty by considering the training dynamic. In self-ensembling learning, the model ensemble is obtained using an exponential running average of model snapshots. In the semi-supervised setup, an unsupervised loss term ensures the model's output remains consistent with different perturbations. For example, given a set of labeled or unlabeled samples, the unsupervised loss enforces the model's output at the current epoch to be similar to the output aggregated from previous training epochs under different perturbations. The aggregated outputs can be obtained using the moving average of the model's output Laine & Aila (2017) or the model's weights Tarvainen & Valpola (2017) throughout the training. In the first phase of FairDSR, we use a similar approach to train the attribute classifier with a reliable uncertainty measure in the data without demographic information. In addition to supervised loss, we add an unsupervised loss term that ensures the model's uncertainty remains consistent within epochs.

On the other hand, considering the other aspect of our hypothesis, the uncertainty estimated by our method is used to derive a subset of the dataset having higher uncertainty in the demographic information prediction. We demonstrate that using this subset to train a model without fairness constraints can yield fairer outcomes. Furthermore, we show how other uncertainty measures such as conformal prediction (Angelopoulos et al., 2023) can be integrated with our framework while achieving similar performance gains regarding tradeoffs in fairness and accuracy. Our main contributions are summarized as follows:

- We show that the fairness-accuracy tradeoff is suboptimal when using proxy-sensitive attributes instead of true-sensitive attributes. Specifically, when the sensitive attribute is partially available, replacing missing values using data imputation techniques based on k-nearest neighbor (KNN) or DNN models can yield a reasonably fair model but a worse fairness-accuracy tradeoff.

- We propose a simple yet effective framework to improve the tradeoff by exploiting the uncertainty of the sensitive attribute predictions, which we show can play an important role in achieving better accuracy-fairness tradeoffs. We hypothesize that a better fairness-accuracy tradeoff can be achieved when fairness constraints are imposed on samples whose sensitive attribute values can be predicted with low uncertainty. We also show that a model trained without fairness constraints but using data with higher uncertainty in the predictions of sensitive attributes tends to be fairer across different fairness metrics.

- We perform extensive experiments on a wide range of real-world datasets to demonstrate the effectiveness of the proposed framework compared to existing methods. Our results also show that the FairDSR can significantly outperform a model trained with fairness constraints on observed sensitive attributes. Applying our method in settings where demographic information is fully available can yield better fairness-accuracy tradeoffs.

## 2 Related Work.

Various metrics have been proposed in the literature to measure unfairness in classification, as well as numerous methods to enforce fairness as per these metrics. The most popular fairness metrics include demographic parity (Dwork et al., 2012), equalized odds, and equal opportunity (Hardt et al., 2016). Demographic parity enforces the models' positive outcome to be independent of the sensitive attributes, while equalized odds aim at equalizing models' true positive and false positive rates across different demographic groups. Fairness-enhancing methods are categorized into three groups: pre-processing (Zemel et al., 2013; Kamiran & Calders, 2012), in-processing (Agarwal et al., 2018; Zhang et al., 2018), and post-processing (Hardt et al., 2016), depending on whether the fairness constraint is enforced before, during, or after model training respectively. However, enforcing these fairness notions often requires access to demographic information. There are fairness notions that do not require demographic information to be achieved, such as the *Rawlsian Max-Min* fairness notion (Rawls, 2020), which aims at maximizing the utility of the worst-case (unknown) group (Hashimoto et al., 2018; Lahoti et al., 2020; Liu et al., 2021; Levy et al., 2020). Specifically, these methods focus on maximizing the accuracy of the unknown worst-case group. However, they often fall short in effectively targeting the specific disadvantaged demographic groups or improving group fairness metrics (Franke, 2021; Lahoti et al., 2020). In contrast, we aim to achieve group fairness through proxy-attributes using limited demographic information. Recent efforts have explored bias mitigation when demographic information is noisy (Wang et al., 2020; Chen et al., 2022a). Noise can be introduced in the sensitive feature space due to human annotation, privacy mechanisms, or inference (Chen et al., 2022b). Chen et al. (2022a) aims to correct the noise in the sensitive attribute space before using them in fairness-enhancing algorithms. Another line of work focuses on alleviating privacy issues in collecting and using sensitive attributes. This group of methods aims to train fair models under privacy-preservation of the sensitive attributes. They design fair models using privacy-preserving mechanisms such as trusted third party (Veale & Binns, 2017), secure multiparty computation (Kilbertus et al., 2018), and differential privacy (Jagielski et al., 2019).

The most related work includes methods relying on proxy-sensitive attributes to enforce fairness when partial demographic information is available. Gupta et al. (2018) used non-protected features to infer proxy demographic information to replace the unobserved real ones. They showed empirically that enforcing fairness with respect to proxy groups generalizes well to the real protected groups and can be effective in practice. While they focus on post-processing techniques, we are interested in in-processing methods. Coston et al. (2019); Liang et al. (2023) assumed sensitive attribute is available either in a source domain or the target domain and used domain adaptation-like techniques to enforce fairness in the domain with missing sensitive attributes. Moreover, while these methods can improve fairness regarding true sensitive attributes, they result in a worse trade-off fairness accuracy than the method using the true sensitive attributes. Diana et al. (2022) showed that training proxy classifier under multi-accuracy constraints can be a good substitute for the ground truth sensitive attributes when the latter is missing; however, the resulting proxy sensitive attributes also yield worse fairness-accuracy tradeoff. Liang et al. (2023) also considers the uncertainty in sensitive space and proposes to replace uncertain predictions with random sampling from the empirical conditional distribution of the sensitive attributes given the non-sensitive attributes $P(A|X, Y)$. However, the random sampling of sensitive attributes is sensitive to noise in the empirical distribution, which can negatively impact

the fairness accuracy tradeoff. Moreover, this work provides a comprehensible analysis of the impact of uncertainty in sensitive attribute space over the fairness-accuracy tradeoff. On the other hand, Awasthi et al. (2021) proposed an active sampling approach to improve bias assessment using predicted sensitive attributes. While their method focuses on bias assessment using proxy attributes, we want to improve the tradeoff between fairness and accuracy, and bias assessment under missing sensitive attributes is out of the scope of the paper. In sum, related work relying on proxy-sensitive attributes mostly focuses on assessing what level of fairness can be achieved when proxy-sensitive attributes are used (Coston et al., 2019), properties of the sensitive attribute classifier (Diana et al., 2022; Coston et al., 2019), and bias assessment via proxy sensitive features (Awasthi et al., 2021). Our proposed method focuses on reducing the accuracy-fairness tradeoff when proxy attributes are used.

## 3 Problem Setting and Preliminaries.

**Problem formulation.** We consider a dataset $\mathcal{D}_1 = \{\mathcal{X}, \mathcal{Y}\}$ where $\mathcal{X} = \{x_i\}_{i=1}^M$ represents the non-sensitive input feature space and $\mathcal{Y} = \{0, 1\}$ represents the target variable. The goal is to build a classifier, $f : \mathcal{X} \to \mathcal{Y}$, that can predict $\mathcal{Y}$ while ensuring fair outcomes for samples from different demographic groups. However, demographic information of samples in $\mathcal{D}_1$ is unknown. We assume the existance of another dataset $\mathcal{D}_2 = \{\mathcal{X}, \mathcal{A}\}$ sharing the same input feature space as $\mathcal{D}_1$ and for which demographic information is available, i.e., $\mathcal{A} = \{0, 1\}$. We assume binary demographic groups for simplicity. Therefore, the dataset $\mathcal{D}_1$ contains label information and $\mathcal{D}_2$ contains demographic information. Our goal is to leverage $\mathcal{D}_2$ to train an attribute classifier $h : \mathcal{X} \to \mathcal{A}$ that can serve as a proxy to the sensitive attributes for samples in $\mathcal{D}_1$, for which a fairness metric can be enforced in a way to improve fairness with respect to the true sensitive attributes. Attribute classifiers have been used in health (Brown et al., 2016; Fremont et al., 2005) and finance (Zhang, 2018; Silva et al., 2019) to infer missing sensitive attributes, in particular when users or patients self-report their protected information. To be able to estimate the true disparities in the label classifier $f$, we assume there exists a small set of samples drawn from the joint distribution $\mathcal{X} \times \mathcal{Y} \times \mathcal{A}$, i.e., samples that jointly have label and demographic information. If this subset is not available, one can consider using the active sampling technique proposed by Awasthi et al. (2021) to approximate bias with respect to the predicted sensitive attributes. This estimation is beyond the scope of this work. We aim to effectively assess fairness without being overly concerned about bias overestimation or underestimation.

Reducing the tradeoff between fairness and accuracy is a significant challenge within the fair machine-learning community (Dutta et al., 2020; Kenfack et al., 2021). Our primary goal is to design a method that effectively leverages proxy features to achieve similar or better fairness-accuracy tradeoffs compared to settings where the true sensitive attributes are available. To this end, we considered a different range of fairness metrics and fairness-enhancing techniques.

**Fairness Metrics.** In this work, we consider three popular group fairness metrics: demographic parity (Dwork et al., 2012), equalized odds, and equal opportunity (Hardt et al., 2016). These metrics aim to equalize the model's performance across different demographic groups; we provide more details in Appendix B.

**Fairness Mechanism.** We focus on in-processing techniques to improve the models' fairness. These methods introduce constraints in the classification problem to satisfy a given fairness metric. Our study focuses on state-of-the-art techniques in this category, i.e., exponentiated gradient (Agarwal et al., 2018) and adversarial debiasing (Zhang et al., 2018). We considered these methods as they allow better control over fairness and accuracy. In general, the optimization problem contains a parameter $\lambda$ that controls the balance between fairness and accuracy, i.e., a higher value of $\lambda$ would force the model to achieve higher fairness (respectively lower accuracy) while a smaller yields higher accuracy (respectively lower fairness). We aim to design a sensitive attribute predictor that achieves a better fairness-accuracy tradeoff, i.e., for the same value of $\lambda$ build a model that provides higher accuracy and lower unfairness.

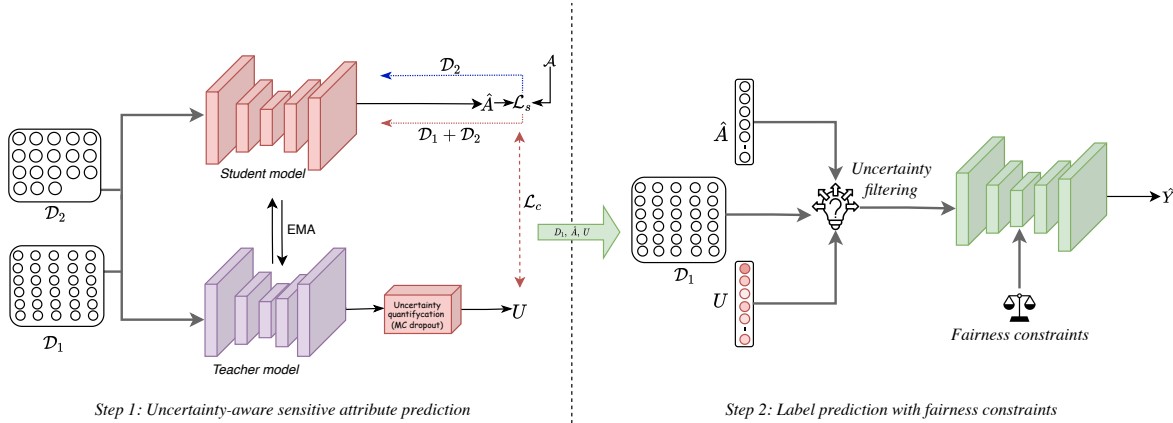

*Step 1: Uncertainty-aware sensitive attribute prediction*          *Step 2: Label prediction with fairness constraints*

Figure 1: **Overview of FairDSR.** Our framework consists of two steps. In the first step (left), the dataset $\mathcal{D}_2$ is used to train the attribute classifier for the student-teacher framework. The first step produces proxy-sensitive attributes ($h(X) = \hat{A}$) and the uncertainty of their predictions ($U$). In the second step (right), the fair model is trained using only samples with reliable proxy-sensitive attributes. These samples are selected based on a defined threshold of their uncertainties.

## 4    Method

This section presents the methodology and all components involved in the framework FairDSR. Figure 1 presents an overview of the stages in our framework and the interactions between and within each stage. The first stage consists of *training the attribute classifier* with uncertainty awareness. This step outputs for each sample with missing sensitive attribute, its predicted sensitive attribute (proxy), and the prediction's uncertainty. In the second stage, *the label classifier is trained with fairness constraints enforced using the predicted sensitive attributes*. To validate our hypothesis, fairness is enforced only on samples with the lowest uncertainty in the sensitive attribute prediction, i.e., samples with an uncertainty lower than a predefined uncertainty threshold $H$.

### 4.1    Uncertainty-Aware Attribute Prediction

We build the sensitive attribute classifier using a semi-supervised learning approach that accounts for the uncertainty of the predictions of samples with missing sensitive attributes, similar to Yu et al. (2019); Tarvainen & Valpola (2017); Laine & Aila (2017). Motivated by the uncertainty estimation in Bayesian networks, we estimate the uncertainty with the Monte Carlo Dropout (Gal & Ghahramani, 2016). However, measuring the uncertainty of the predictions using the model in a given training step (e.g., the last epoch) might result in an unreliable uncertainty measure due to different perturbations, such as network dropout, input noise, and data ordering (He et al., 2023). In addition, we are interested in designing an attribute classifier that minimizes the uncertainty of the predictions in the data with missing sensitive attributes.

To overcome this challenge, we adopt self-ensemble learning to ensure consistent uncertainty measurement by maintaining the moving average of the attribute classifier's weights during the training (Tarvainen & Valpola, 2017). More specifically, the model maintains a moving average of itself during the training to provide a stable learning signal. We refer to the model at the current training epoch as the *student model* and its moving average (self-ensembling) from previous epochs as the *teacher model*.

**Student Model.**    The student model is implemented as a neural network and is trained on $\mathcal{D}_2$ (samples with sensitive attributes) to predict sensitive attributes. Under a semi-supervised setup, the attribute classifier is optimized to minimize a double loss: the *classification loss* ($\mathcal{L}_s$), i.e., the cross-entropy loss, and the *consistency loss* ($\mathcal{L}_c$) (Yu et al., 2019). The consistency loss (or unsupervised loss) forces the student model to focus on samples with low uncertainty in the sensitive attributes guided by the uncertainty estimation

from the teacher model. This loss is defined as the mean squared difference between the teacher's and student outputs (logits) on samples whose uncertainty does not exceed a threshold $R$.

Overall, the attribute classifier is trained to minimize the following loss:

$$\min_{f \in \mathcal{F}} \mathop{\mathbb{E}}_{(x,a) \sim \mathcal{D}_2 \times \mathcal{A}} \mathcal{L}_s(f(x), a) + \lambda \mathop{\mathbb{E}}_{x \sim \mathcal{D}_1 + \mathcal{D}_2} \mathcal{L}_c(f(x), h(x)) \tag{1}$$

where $f(\cdot)$ is the student model, $h(\cdot)$ the teacher model, and $\lambda$ a parameter controlling the consistency loss. The empirical loss minimized is defined by the following equations for classification ($\mathcal{L}_s$) and consistency loss $\mathcal{L}_c$:

$$\mathcal{L}_s = \frac{1}{|\mathcal{D}_2|} \sum_{x,a \in \mathcal{D}_2, A} a \cdot \log(f(x)) + (1-a) \cdot \log(1 - f(x)) \tag{2}$$

$$\mathcal{L}_c = \frac{1}{|\mathcal{D}_2| + |\mathcal{D}_1|} \sum_{x | u_x \leq R} \| f(x) - h(x) \|^2 \tag{3}$$

The consistency loss is applied only on samples, $x$, whose uncertainty, $u_x$, is lower than the threshold $R$. Following Srivastava et al. (2014); Baldi & Sadowski (2013), $R$ and $\lambda$ are updated using a Gaussian warmup function to prevent the model from diverging at the beginning of the training. The motivation behind the consistency loss is the focus on our primary goal for the attribute classifier, which is to find the missing sensitive attributes in $\mathcal{D}_1$ with low uncertainty. In the experiments (Section 5.2.3), we show that training the model without consistency loss results in worse fairness-accuracy tradeoffs.

**Teacher Model.** The teacher model maintains the moving average of the student's weights and is used for uncertainty estimation. Specifically, the teacher weights are updated within each training epoch, $t$, using the exponential moving average (EMA) of student weights as follows:

$$\omega_t = \alpha \omega_{t-1} + (1 - \alpha)\theta, \tag{4}$$

where $\theta$ and $\omega$ denote the respective weights of student and teacher and $\alpha$ controls the moving decay. The use of EMA to update the teacher model is motivated by previous studies (Laine & Aila, 2017; Yu et al., 2019) that have shown that averaging model parameters at different training epochs can provide more reliable predictions than using the most recent model weights in the last epoch. The teacher model gets as input both labeled and unlabeled samples ($\mathcal{D}_1$ and $\mathcal{D}_2$) and computes the uncertainty of their predictions using Monte Carlo (MC) dropout (Gal & Ghahramani, 2016).

**Uncertainty Estimation.** MC dropout is an approximation of a Bayesian neural network widely used to interpret the parameters of neural networks (Abdar et al., 2021). MC dropout can effectively capture various sources of uncertainties (Aleatoric and Epistemic uncertainty). It uses dropout at test time to compute prediction uncertainty from different sub-networks that can be derived from the whole original neural network. Dropout is generally used to improve the generalization of DNNs. During training, the dropout layer randomly removes a unit with probability $p$. Therefore, each forward and backpropagation pass is done on a different model (sub-network), forming an ensemble of models that are aggregated together to form a final model with lower variance (Srivastava et al., 2014; Baldi & Sadowski, 2013). The uncertainty of each sample is computed using $T$ stochastic forward passes on the teacher model to output $T$ independent and identically distributed predictions, i.e., $\{h_1(x), h_2(x), \cdots, h_T(x)\}$. The softmax probability of the output set is calculated. The uncertainty of the prediction ($u_x$) is quantified using the resulting entropy: $u_x = -\sum_a p_a(x) \log(p_a(x))$, where $p_a(x)$ is the probability that sample $x$ belongs to demographic group $a$ estimated over $T$ stochastic forward passes, i.e., $p_a(x) = \frac{1}{T} \sum_{t=1}^{T} h_t^a(x)$. In the experiments (Section 5.2.3), we compare with another uncertainty measure based on the confidence interval of the prediction probability and show the limitation of this measure in identifying highly uncertain samples. Furthermore, we also show that using reliable uncertainty measures such as conformal prediction (Angelopoulos et al., 2023) can also improve the fairness-accuracy tradeoff.

### 4.2 Enforcing Fairness w.r.t Reliable Proxy Sensitive Attributes

After the first phase, the attribute classifier can produce for every sample in $\mathcal{D}_1$, i.e., samples with missing sensitive attributes, their predicted sensitive attribute (proxy) $\hat{A} = \{h(x_i)_{x_i \in \mathcal{D}_1}\}$, and the uncertainty of the prediction $U = \{u_{x_i}\}_{x_i \in \mathcal{D}_1}$. To validate our hypothesis, we define a confidence threshold $H$ for samples used to train the label classifier with fairness constraints, i.e., the label classifier with fairness constraints is trained on a subset $\mathcal{D}_1' \subset \mathcal{D}_1$ defined as follows:

$$\mathcal{D}_1' = \{(x, y, f(x)) | u_x \leq H\} \tag{5}$$

Note that the threshold $R$ in the previous step is only used to train the attributes classifier while ensuring the student and teacher remain consistent for samples with uncertainty lower than $R$. The threshold $H$ in this step is used at test time to select samples with low uncertainty, and its value can be tuned over a validation set. The hypothesis of enforcing fairness on samples whose sensitive attributes are reliably predicted stems from the fact that the model can confidently distinguish these samples based on their sensitive attributes in the latent space. In contrast, the label classifier is inherently fairer if an attribute classifier cannot reliably predict sensitive attributes from training data (Kenfack et al., 2023). We further support this in section 5.2 by comparing the sensitive attribute uncertainty in the New Adult dataset (Ding et al., 2021) and the old version of the dataset (Asuncion & Newman, 2007). The fairness constraints on samples with unreliable sensitive attributes could push the model's decision boundary in ways that penalize accuracy and/or fairness. We support these arguments in the experiments. In addition to this method, we considered two other approaches built around our hypothesis: the weighted approach— FairDSR (weighted) and the uncertain approach— FairDSR (uncertain)

**FairDSR (weighted)**. Given that downsampling can reduce accuracy when uncertainty is low, instead of pruning out samples based on the uncertainty of their sensitive attributes, we considered a training process in the second step that enforces fairness constraints on all samples but weighted proportionally to the uncertainty of their sensitive attributes, i.e., samples with less uncertainty receive higher weights. We show in the experiments that this weighted approach can better preserve accuracy.

**FairDSR (uncertain)**. Considering the ethical risks of inferring missing sensitive information, we also considered a variant in the second step where the model is trained without fairness constraints, i.e., without using sensitive information. Instead, we train the model using samples with higher uncertainty in their sensitive attributes for a given uncertainty threshold. We show in the experiments that this variant can significantly improve fairness, although no fairness constraints are applied.

## 5 Experiments

In this section, we demonstrate the effectiveness of our framework on five datasets and compare it to different baselines.

### 5.1 Experimental Setup

**Datasets.** We validate our method on five real-world benchmarks widely used for bias assessment: Adult Income (Asuncion & Newman, 2007)[1], Compas (Jeff et al., 2016), Law school (LSAC), CelebA (Liu et al., 2018) (Wightman, 1998), and the New Adult (Ding et al., 2021) dataset. We use 20% of each dataset as the group-labeled dataset ($\mathcal{D}_2$) and 80% as the dataset without sensitive attributes ($\mathcal{D}_1$). Supplementary C.1 shows that our results hold even with a smaller group-labeled ratio, e.g., 5%. More details about the datasets appear in Supplementary B.2.

**Attribute classifier.** The student and teacher models were implemented as feed-forward Multi-layer Perceptrons (MLPs) with Pytorch (Paszke et al., 2019), and the loss function 1 is minimized using the Adam optimizer (Kingma & Ba, 2014) with learning rate 0.001 and batch size 256. Following Yu et al. (2019); Laine & Aila (2017), we used $\alpha = 0.99$ for the EMA parameter for updating the teacher weights using the student's

---

[1]https://archive.ics.uci.edu/ml/datasets/Adult

weights across epochs. The uncertainty threshold is finetuned over the interval $[0.1, 0.7]$ using 10% of the training data. The best-performing threshold is used for the second step's threshold to obtain $\mathcal{D}'_1$. The uncertainty threshold that achieved the best results are 0.30, 0.60, 0.66, and 0.45 for the Adult, Compas, LSAC, and CelebA datasets, respectively.

**Baselines.** For fairness-enhancing mechanisms, we considered the Fairlean (Bird et al., 2020) implementation of the exponentiated gradient (Agarwal et al., 2018). We considered the three variants of our approach described in section 4.2: FairDSR (weighted), FairDSR (uncertain), and our base method, FairDSR (certain).

For comparison, we considered methods aiming to improve fairness without (full) demographic information. We compare with the following methods:

- FairRF (Zhao et al., 2022): This method assumes that non-sensitive features correlating with sensitive attributes are known. It leverages these related features to improve fairness w.r.t the unknown sensitive attributes.
- FairDA (Liang et al., 2023): Similar to our setting, this method assumes the sensitive information is available in a *source domain* (dataset $\mathcal{D}_2$ in our setting). It uses a domain adaptation-based approach to transfer demographic information from the source domain to improve fairness in the target using an adversarial approach.
- CGL (Jung et al., 2022): With partial access to sensitive attributes, this method also uses an attribute classifier and replaces uncertain predictions with random sampling from the empirical conditional distribution of the sensitive attributes given the non-sensitive attributes $P(A|X, Y)$.
- ARL (Lahoti et al., 2020): The method uses an adversarial approach to upweight samples in regions hard to learn, i.e., regions where the model makes the most mistakes.
- Distributionally Robust Optimization (DRO) (Hashimoto et al., 2018): It optimizes for the worst-case distribution around the empirical distribution. Similar to ARL, the goal is to improve the accuracy of the worst-case group.
- CVarDRO (Levy et al., 2020): It is an improved variant of DRO.
- KSMOTE (Yan et al., 2020) performs clustering to obtain pseudo groups and use them as substitutes to oversample the minority groups.

For each method considered for comparison, we used the code provided by the authors[2] along with the recommended hyperparameters. We considered two *vanilla* baselines: a baseline where the model is trained without fairness constraints (Vanilla (without fairness)) and a baseline model trained with fairness constraints over the true sensitive attributes (Vanilla (with fairness)). We also considered baselines trained with fairness constraints directly using predicted sensitive attributes obtained by data imputation using our student network (Proxy-DNN) and imputation using K-nearest neighbor (Proxy-KNN).

For comparison, in addition to the accuracy, we consider the three fairness metrics described in the appendix B, i.e., equalized odds ($\Delta_{\text{EOD}}$), equal opportunity ($\Delta_{\text{EOP}}$), and demographic parity ($\Delta_{\text{DP}}$). All the baselines are trained on 70% of $\mathcal{D}_1$, and fairness and accuracy are evaluated on the 30% as the test set. We assume the sensitive attribute is observed in the test set to report the true fairness violation. We trained each baseline 7 times and averaged the results. We use logistic regression[3] as the base classifier for all the baselines and train each baseline to achieve minimal fairness violation.

## 5.2 Results and Discussion

### 5.2.1 Uncertainty of the sensitive attribute and fairness.

We first analyze the relation between the sensitive attribute's uncertainty and the fairness of downstream models. Table 1 showcases the average uncertainty of the sensitive attribute prediction estimated by our

---

[2]We implemented FairDA and reproduced using the instructions in the paper (Liang et al., 2023).
[3]Appendix C.2 shows a comparison with an MLP model.

| Dataset | Mean uncertainty ($\downarrow$) | Accuracy sensitive attribute ($\uparrow$) | $\Delta_{DP}$ | $\Delta_{EOD}$ | $\Delta_{EOP}$ |
|---|---|---|---|---|---|
| Adult | 0.15 | 85% | 0.18 | 0.20 | 0.13 |
| New Adult | 0.42 | 68% | 0.06 | 0.05 | 0.04 |
| Compas | 0.39 | 72% | 0.28 | 0.32 | 0.32 |
| CelebA | 0.21 | 83% | 0.17 | 0.19 | 0.19 |
| LSAC | 0.66 | 55% | 0.014 | 0.005 | 0.049 |

Table 1: Average uncertainty of the attribute classifier and fairness of a label classifier on the dataset with missing sensitive attributes.

| Method | Accuracy | $\Delta_{DP}$ | $\Delta_{EOP}$ | $\Delta_{EOD}$ |
|---|---|---|---|---|
| Vanilla (without fairness) | $0.851 \pm 0.008$ | $0.171 \pm 0.004$ | $0.088 \pm 0.033$ | $0.091 \pm 0.030$ |
| Vanilla (with fairness) | $0.829 \pm 0.002$ | $0.005 \pm 0.004$ | $0.021 \pm 0.014$ | $0.017 \pm 0.007$ |
| Proxy-DNN | $0.829 \pm 0.002$ | $0.009 \pm 0.005$ | $0.024 \pm 0.014$ | $0.019 \pm 0.017$ |
| FairRF | $0.838 \pm 0.002$ | $0.162 \pm 0.015$ | $0.063 \pm 0.027$ | $0.072 \pm 0.019$ |
| FairDA | $0.809 \pm 0.009$ | $0.087 \pm 0.028$ | $0.071 \pm 0.046$ | $0.078 \pm 0.039$ |
| CGL | $0.834 \pm 0.002$ | $0.009 \pm 0.006$ | $0.027 \pm 0.020$ | $0.026 \pm 0.016$ |
| ARL | $\mathbf{0.850} \pm 0.002$ | $0.173 \pm 0.013$ | $0.028 \pm 0.090$ | $0.097 \pm 0.031$ |
| CVarDRO | $0.820 \pm 0.012$ | $0.200 \pm 0.005$ | $0.160 \pm 0.030$ | $0.100 \pm 0.027$ |
| KSMOTE | $0.814 \pm 0.003$ | $0.302 \pm 0.007$ | $0.160 \pm 0.021$ | $0.196 \pm 0.003$ |
| DRO | $0.823 \pm 0.003$ | $0.184 \pm 0.042$ | $0.092 \pm 0.041$ | $0.105 \pm 0.041$ |
| FairDSR (weighted) | $0.846 \pm 0.003$ | $0.032 \pm 0.013$ | $0.050 \pm 0.033$ | $0.027 \pm 0.019$ |
| FairDSR (uncertain) | $0.825 \pm 0.013$ | $0.106 \pm 0.036$ | $0.065 \pm 0.047$ | $0.068 \pm 0.032$ |
| FairDSR (certain) | $0.830 \pm 0.004$ | $\mathbf{0.007} \pm 0.005$ | $\mathbf{0.015} \pm 0.010$ | $\mathbf{0.018} \pm 0.016$ |

Table 2: **Results on the Adult dataset.** Bolded values represent the best-performing baselines among the fairness-enhancing methods without (full) demographic information. All the baselines are trained on the dataset without the sensitive attributes ($\mathcal{D}_1$). Each experiment is conducted 7 times, and the fairness and accuracy are averaged.

method and different fairness measures of a logistic regression model trained without fairness constraints on the dataset $\mathcal{D}_1$. These results show the correlation between the uncertainty of the sensitive attribute prediction and the fairness of the model. For example, we observe that the uncertainty in the Adult dataset is lower than in the New Adult dataset, while the unfairness in the Adult dataset is higher. On the other hand, the LSAC dataset has the highest uncertainty of the sensitive attribute (0.66), and the model without fairness constraints has the smallest fairness violation across all fairness metrics. In sum, we can observe that unfairness is higher for datasets with a lower uncertainty in the sensitive attributes, e.g., the Adult and CelebA datasets. These results support our hypothesis that a model can hardly discriminate against samples with uncertain demographic information. Furthermore, we investigate the relation between the level of uncertainty of the sensitive attributes in the training data and the fairness of downstream classifiers trained without fairness constraints. Our study reveals that a model without fairness constraints tends to be fairer as the uncertainty of the sensitive attribute in the training data increases.

Specifically, we trained different classifiers (Logistic Regression and Random Forest) without fairness constraints but using training data with higher uncertainty in the sensitive attributes. For different uncertainty thresholds $H \in \{0.0, 0.1, ..., 0.6\}$, we prune out samples whose uncertainty is lower than $H$ and train the model without fairness constraints using the remaining training data, i.e., $\{(x, y) \in D_1 | u_x \geq H\}$, where $u_x$ is the estimated uncertainty provided by our method. In particular, when $H = 0$, all the data points are used for the training, and in other cases, we only maintain samples with uncertainty higher than $H$, i.e., the model is trained on samples with more uncertain sensitive attributes. We train the model seven times with different random seeds for each uncertainty threshold and report fairness and accuracy in the testing set containing sensitive attributes.

| Method | Accuracy | $\Delta_{\text{DP}}$ | $\Delta_{\text{EOP}}$ | $\Delta_{\text{EOD}}$ |
|---|---|---|---|---|
| Vanilla (without fairness) | $0.681 \pm 0.011$ | $0.285 \pm 0.026$ | $0.325 \pm 0.029$ | $0.325 \pm 0.029$ |
| Vanilla (with fairness) | $0.634 \pm 0.009$ | $0.032 \pm 0.011$ | $0.039 \pm 0.024$ | $0.041 \pm 0.016$ |
| Proxy-DNN | $0.634 \pm 0.005$ | $0.049 \pm 0.008$ | $0.099 \pm 0.036$ | $0.080 \pm 0.022$ |
| FairRF | $0.669 \pm 0.001$ | $0.289 \pm 0.003$ | $0.319 \pm 0.004$ | $0.319 \pm 0.004$ |
| FairDA | $0.668 \pm 0.019$ | $0.229 \pm 0.018$ | $0.265 \pm 0.024$ | $0.265 \pm 0.024$ |
| CGL | $0.612 \pm 0.019$ | $0.032 \pm 0.014$ | $0.065 \pm 0.019$ | $\mathbf{0.065} \pm 0.027$ |
| ARL | $0.672 \pm 0.009$ | $0.290 \pm 0.016$ | $0.310 \pm 0.010$ | $0.320 \pm 0.010$ |
| CVarDRO | $0.668 \pm 0.008$ | $0.279 \pm 0.018$ | $0.300 \pm 0.010$ | $0.287 \pm 0.015$ |
| KSMOTE | $0.670 \pm 0.012$ | $0.286 \pm 0.028$ | $0.321 \pm 0.028$ | $0.321 \pm 0.028$ |
| DRO | $0.672 \pm 0.010$ | $0.282 \pm 0.026$ | $0.296 \pm 0.017$ | $0.296 \pm 0.017$ |
| FairDSR (weighted) | $0.672 \pm 0.009$ | $\mathbf{0.027} \pm 0.016$ | $0.069 \pm 0.038$ | $0.083 \pm 0.035$ |
| FairDSR (uncertain) | $0.671 \pm 0.009$ | $0.272 \pm 0.016$ | $0.300 \pm 0.039$ | $0.300 \pm 0.034$ |
| FairDSR (certain) | $\mathbf{0.676} \pm 0.009$ | $0.085 \pm 0.016$ | $\mathbf{0.067} \pm 0.039$ | $0.074 \pm 0.034$ |

Table 3: Results on the Compas dataset.

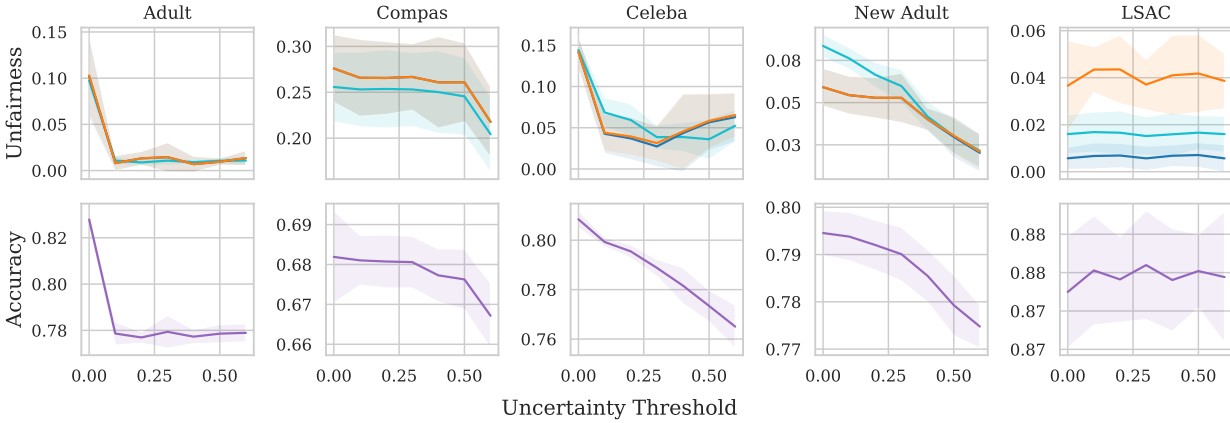

Figure 2: Training Random Forest classifiers without fairness constraints using samples with high uncertainty of sensitive attribute predictions. For each uncertainty threshold $H$, the model is trained on samples with uncertainty $\geq H$. The training is done seven times, and the average fairness (first row) and accuracy (second row) are reported. Shaded represents the standard deviation.

Figure 2 shows the fairness and accuracy of a Random Forest classifier trained without fairness constraints. In the figure, each column represents the results on each dataset, and the first and the second rows provide the plots of fairness and accuracy for different uncertainty thresholds, respectively. Across different datasets, the results show that unfairness decreases as the uncertainty threshold increases. We observe that the improvement in fairness is consistent for different fairness metrics considered, i.e., demographic parity, equal opportunity, and equalized odd. We also observe a decrease in the accuracy, which is justified by reduced dataset size and improved fairness (tradeoff). On the LSAC dataset, fairness and accuracy remain almost constant as the average uncertainty in predicting the sensitive attribute on this dataset is 0.66, i.e., most of the samples already have the highest uncertainty. We observed similar results for a Logistic Regression classifier presented in the Supplementary (Figure 8). However, this method incurs a higher drop in accuracy and does not necessarily guarantee that an adversary cannot reconstruct the sensitive attributes from the trained model (Ferry et al., 2023).

### 5.2.2 Fairness-accuracy tradeoffs.

Table 2, and 3 show the effectiveness of the FairDSR compared to other baselines on the Adult and Compas datasets, respectively. Results for the CelebA and LSAC appear in Appendix (Table 6 and 7, respectively).

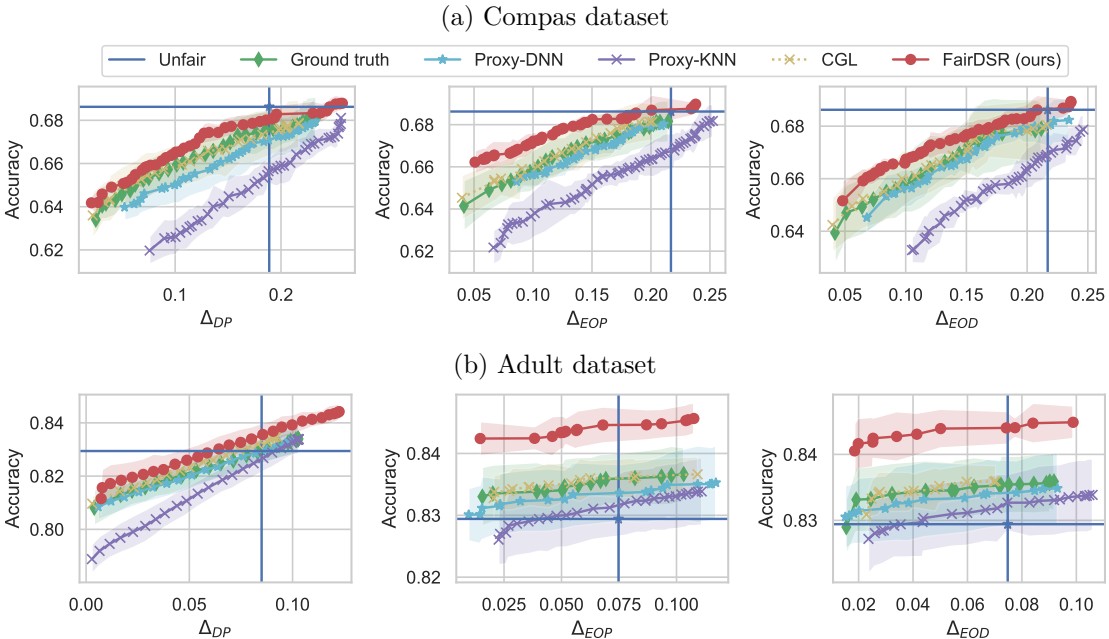

Figure 3: Accuracy-fairness tradeoffs for various fairness metrics ($\Delta_{\mathrm{DP}}$, $\Delta_{\mathrm{EOP}}$, $\Delta_{\mathrm{EOD}}$) and proxy sensitive attributes. The top-left is the best (highest accuracy with the lowest unfairness). Curves are created by sweeping a range of fairness coefficients $\lambda$, taking the median of 7 runs per $\lambda$, and computing the Pareto front. The exponentiated gradient is the fairness mechanism with Random Forests as the base classifier. The standard deviations are shaded in the figures.

It is important to note that methods aiming to improve worst-case group accuracy (ARL, DRO, CVarDRO) do not necessarily improve fairness in terms of demographic party or equalized odds. In particular, results across different datasets show that ARL can improve the Equal Opportunity metric but fails to improve demographic parity. It also yields the most accurate classifier as this method does not have a tradeoff with accuracy. On the other hand, FairDA (Liang et al., 2023) and CGL (Jung et al., 2022), which also exploit limited demographic information, show an improvement in fairness compared to other baselines. However, it incurs a higher drop in accuracy while our method using fairness constraints on samples with reliable sensitive attributes significantly outperforms them across all datasets. In other words, the results show that our method with fairness constraints on samples with reliable sensitive attributes provides Pareto dominant points in terms of fairness and accuracy.

On the other hand, the variant using a model trained without fairness constraints (without using sensitive attributes) provides better fairness-accuracy tradeoffs compared to other baselines on the Adult and the CelebA datasets while providing comparable results on datasets with higher uncertainty (LSAC and Compas). For example, the LSAC dataset has an average uncertainty of 0.66, meaning most samples already have uncertain sensitive information, and the unfairness is already low. As no fairness constraints are enforced in FairDSR (uncertain), it less impacts fairness and accuracy as most data samples are preserved due to high overall uncertainty. On the other hand, we observe that FairDSR (weighted) can outperform other baselines and provide better accuracy than FairDSR (certain) at the cost of higher unfairness. The improved accuracy is explained by the use of all data points in the weighted approach. At the same time, FairDSR (certain) can strengthen fairness when only samples with reliable sensitive attributes are used. FairDSR (certain) also provides better Pareto dominant points than CGL and FairDA. While CGL, which also accounts for the uncertainty of the sensitive attributes, can outperform other baselines, especially those directly using predicted proxy-sensitive attributes. These results support our hypothesis that using data points with certain sensitive attributes is more beneficial to fairness and accuracy. Figure 3 shows that imputation methods can yield reasonably good fairness-accuracy tradeoffs on the Adult and Compas datasets but are suboptimal

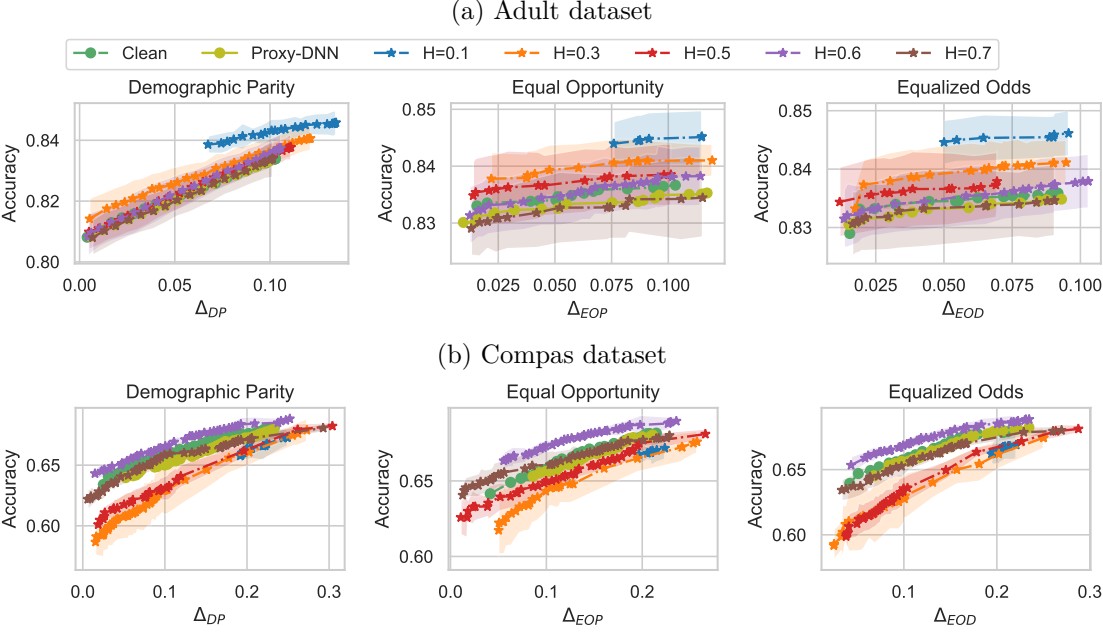

Figure 4: The impact of the uncertainty threshold $H$ on the fairness-accuracy tradeoff for (a) Adult and (b) Compas datasets.

compared to the model using the true sensitive attributes. We observed similar results with different base classifiers (logistic regression and gradient-boosted trees) and fairness mechanisms. These results appear Appendix C.3.

### 5.2.3 Ablation experiments

**Impact of the uncertainty threshold.** Figure 4 showcases the impact of the uncertainty threshold on the fairness-accuracy threshold. When the feature space encodes much information about the sensitive attribute as in the Adult dataset (Figure 4a) with 85% accuracy of predicting the sensitive attributes, the results show that the more we enforce fairness w.r.t. samples with the lower uncertainty, the better the fairness-accuracy tradeoffs. In this regime, enforcing unfairness helps the model maintain a better accuracy level (Figure 4a). In contrast, in a low bias regime, i.e., when the feature space does not encode enough information about the sensitive attributes, such as on the Compas and the New Adult dataset, the model achieves better fairness-accuracy tradeoffs when a higher uncertainty threshold is used. In this regime, most of the samples have higher uncertainty in the sensitive attribute prediction, and fairness violation is smaller (see Table 1), as can be observed in Figure 4b, the use of a lower uncertainty threshold leads to a decrease in accuracy while fairness is improved. We observe similar results in the New Adult, CelebA, and LSAC datasets (Fig 13 in Supplementary). The accuracy drops since more and more samples were pruned out from the datasets, and this suggests that the feature space is more informative for the target task than the demographic information. In the appendix (D), we show that while under-represented demographic groups can have higher uncertainty on average than well-represented groups, minority groups are still consistently represented when a lower threshold is used.

**Importance of the consistency loss.** In the proposed framework, we use trained attribute classifier with a double loss: the cross-entropy loss (supervised loss) and the consistency loss (unsupervised loss). The parameter $\lambda$ controlling the consistency loss is updated using a Gaussian ramp-up function starting from zero at the beginning of the training. The consistency loss pushes the student to focus on data points with highly certain sensitive attributes. Without this second loss term, the student model minimizes the classification error without explicit consideration of the uncertainty in the dataset without sensitive attributes.

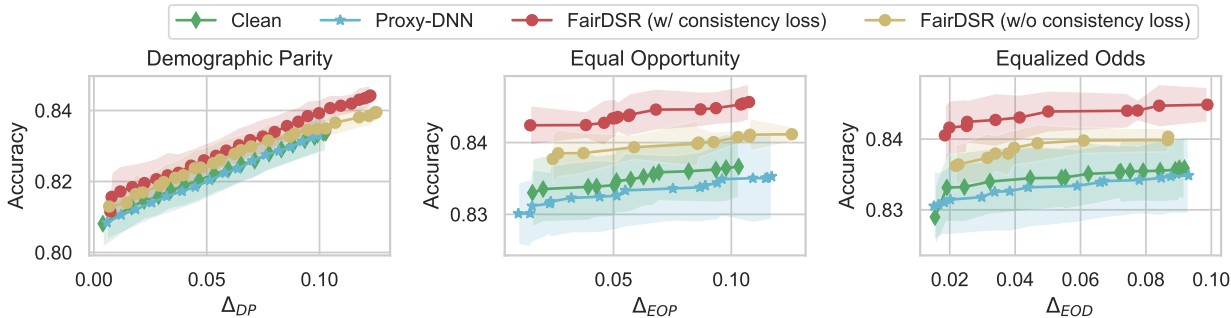

Figure 5: **Consistency loss study on the Adult dataset.** The predicted sensitive attributes are obtained using our student model with and without consistency loss.

To demonstrate the importance of consistency loss (and thus the importance of self-ensembling with the teacher), we experiment with the student model trained without the consistency loss, i.e., we set $\lambda = 0$ in Equation 1. We applied the same procedure of the second step of FairDSR (certain) to derive data points with sensitive attributes predicted with low uncertainty.

Figure 5 showcases the Pareto front of our FairDSR trained without (w/) and without (w/o) the consistency loss. The base classifier is Random Forest with exponentiated gradient as the fairness mechanism. The Figure also shows the Pareto front of the model trained with the ground truth sensitive attribute and the predicted sensitive attribute (Proxy-DNN). We observe that training the model with the consistency loss provides Pareto dominant points across different fairness metrics. This demonstrates the benefit of the consistency loss in effectively identifying samples with sensitive attributes predicted with low uncertainty in the dataset without demographics. We also observed that the mean uncertainty for the model without the consistency loss is around $0.19 \pm 0.08$, while the model that uses the consistency loss is $0.15 \pm 0.12$. This suggests that the consistency loss effectively enforces the student model focus on data points with low uncertainty. Moreover, we also observe that the model without the consistency loss still outperforms the model that directly uses the sensitive attributes (Proxy-DNN) and the model that uses true sensitive attributes. This demonstrates that our hypothesis still holds even if the uncertainty measure is less effective, and a better uncertainty measure can provide even better results.

**Experiment with uncertainty measure based on confidence intervals.** We adopted Monte Carlo dropout because it efficiently captures various sources of uncertainty, both in the data and the model. Another classic approach to measuring uncertainty is the model's prediction confidence. In this experiment, we evaluate the effect of using confidence intervals as the uncertainty measure on fairness-accuracy tradeoffs. Given a trained attribute classifier, recall we obtain the predicted group label by *thresholding* the prediction probability, i.e., $f(x) = \mathbb{1}(P(\hat{A} = a | X = x) \geq 0.5)$.

We construct the confidence interval using a threshold $\tau \in [0.5, 1]$, such that samples with higher uncertainty have their prediction probability closer to $0.5$. A given data point $x$ is in the high uncertainty set if $1 - \tau < P(\hat{A} = a | X = x) < \tau$, while $x$ is in the low uncertainty set otherwise. More specifically, in this experiment, we derive the subset of data points with low uncertainty of the sensitive attribute prediction as follows :

$$(x, y, f(x)) \in \mathcal{D}_1' \text{ if } \begin{cases} P(\hat{A} = a | X = x) \in [0, 1 - \tau], \\ \qquad \text{or} \qquad\qquad\qquad \forall a \in \{0, 1\} \\ P(\hat{A} = a | X = x) \in [\tau, 1], \end{cases} \tag{6}$$

We experiment with different confidence thresholds $\tau$ on the Adult dataset. Specifically, for each confidence threshold $\tau \in \{0.6, 0.7, 0.8, 0.9\}$, we trained the Random Forest model using data points with high confidence. Setting the confidence threshold to $0.5$ is similar to using all data points to train the model. As shown

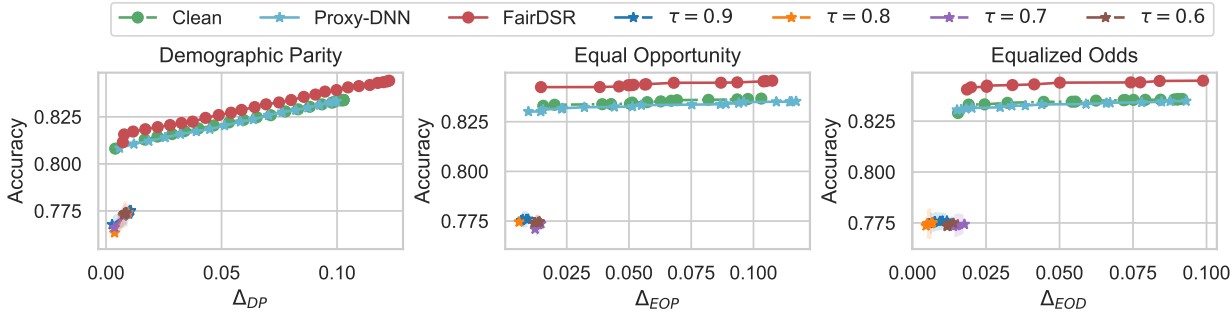

Figure 6: Study of uncertainty measure based on confidence interval.

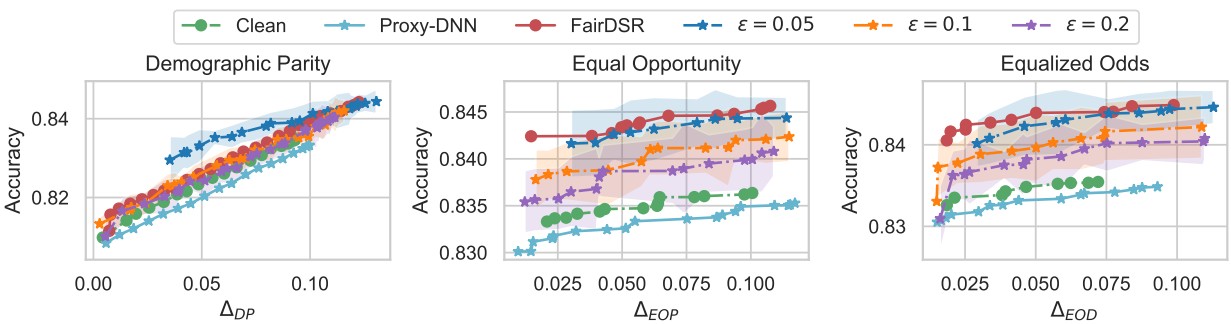

Figure 7: Study of uncertainty measure based on conformal predictions.

in Figure 6, models trained using confidence intervals as uncertainty measures yield worse Pareto points. Specifically, we observe a significant drop in accuracy while fairness performances significantly improve. We also observe that a higher confidence threshold tends to provide better fairness for a similar level of accuracy. While this observation also supports our hypothesis, the drop in accuracy suggests that the confidence interval is less effective in identifying data points with low uncertainty in sensitive attributes. We suspect this results from the unreliability of the model probability due to overconfidence or underconfidence in wrong predictions (Guo et al., 2017). As the confidence interval is obtained using a single model, low-confidence prediction could be solely caused by poor calibration or randomness in the algorithm. We validate this in the next ablation study by calibrating the attribute classifier and using conformal prediction to measure uncertainty.

**Ablation on uncertainty measure based on conformal predictions.** In this experiment, we investigate the validity of our hypothesis under different and reliable uncertainty measures, such as conformal prediction (Shafer & Vovk, 2008). Conformal prediction, a machine learning framework, constructs prediction sets containing possible labels. These sets are guaranteed to include the true label with a probability of $1 - \epsilon$, where $\epsilon \in [0, 1]$ is a user-defined error rate. For instance, setting $\epsilon = 0.1$ ensures the prediction set has at least a 90% chance of containing the correct label. The size of the set reflects the model's confidence in its prediction. Ideally, the set would contain only one label, indicating high confidence. Larger sets suggest greater uncertainty about the predicted label. To construct the prediction set of the sensitive attribute, with the $1 - \epsilon$ guarantee, we implement *split conformal prediction* as described by Angelopoulos et al. (2023), which is the widely-used version of conformal prediction. Split conformal prediction takes the pretrained attribute classifier and calibration set to compute the conformal prediction threshold[4] used to include labels in the prediction set of the test set. We employ the Model Agnostic Prediction Interval Estimator (MAPIE) library to train a conformal classifier and measure the uncertainties in the attribute classier (Cordier et al., 2023). The classifier wraps the original attribute classifier and produces conformal prediction sets with

---

[4]See Angelopoulos et al. (2023) for more details on how prediction sets are constructed

| Sensitive attributes | Coverage $\epsilon$ | Accuracy | $\Delta_{\mathrm{DP}}$ | $\Delta_{\mathrm{EOP}}$ | $\Delta_{\mathrm{EOD}}$ |
|---|---|---|---|---|---|
| Uncertain + Certain | N/A | 0.82 | 0.08 | 0.07 | 0.07 |
| Certain | 0.05 | 0.84 | 0.123 | 0.089 | 0.118 |
| | 0.1 | 0.83 | 0.109 | 0.119 | 0.100 |
| | 0.2 | 0.83 | 0.110 | 0.081 | 0.109 |
| Uncertain | 0.05 | 0.77 | 0.009 | 0.012 | 0.007 |
| | 0.1 | 0.77 | 0.006 | 0.017 | 0.009 |
| | 0.2 | 0.76 | 0.003 | 0.005 | 0.008 |

Table 4: Study of models without fairness constraints using different levels of uncertainty measured with conformal predictions.

the guaranteed marginal coverage rate $1 - \epsilon$. We used 10% of the dataset with sensitive attributes as the calibration set and to train the calibrated attribute classifier.

In the second step of our framework, we generate prediction sets of the sensitive attributes in the dataset where demographic information is missing ($\mathcal{D}_1$). Since the sensitive attributes are binary, there are four possible prediction sets for each data point: $\{0\}$, $\{0, 1\}$, $\{1\}$, and the empty set $\{\varnothing\}$. This means each prediction set for every sample consists of either a single attribute value, both values or no value at all (Gupta et al., 2020). Samples with prediction sets containing exactly one attribute value are grouped into subsets of samples with low uncertainty ($\mathcal{D}_1'$). On the other hand, samples with empty prediction sets or sets containing both attribute values indicate higher uncertainty regarding the sensitive attributes. As we used a score-based method to compute non-conformity scores, the prediction set for uncertainty predictions might be empty if a set size smaller than 1 is necessary to ensure $1 - \epsilon$ coverage (Cordier et al., 2023).

In our experiments on the Adult dataset, we varied $\epsilon$ across values $\{0.05, 0.1, 0.2\}$. This means for $\epsilon = 0.05$, $\epsilon = 0.1$, and $\epsilon = 0.2$, the prediction sets are guaranteed to include the true sensitive attribute with probabilities of 95%, 90% and 80%, respectively. Our model training incorporated fairness constraints specifically on data points where the prediction set contained a single attribute value, indicating higher confidence in the prediction for smaller values of $\epsilon$. Following the previous evaluation, we trained the Random Forest classifier for each $\epsilon$ and sweeping range of fairness coefficients $\lambda$, taking the median of 7 runs and computing the Pareto front. Figure 7 showcases the Pareto front of different models with fairness constraints on samples with greater certainty of the sensitive attributes (samples with single-valued prediction sets) for different values of $\epsilon$. As can be seen in Fig 7, all models trained using samples with low uncertainty in the sensitive attributes achieve better Pareto front than the model using all the predicted sensitive values (Proxy-DNN) and even ground truth sensitive attributes (clean). This demonstrates the benefit of applying fairness constraints over data points with low uncertainty in the sensitive on-fairness accuracy tradeoff. Moreover, the figure shows that smaller values of $\epsilon$ result in better Pareto dominant points, i.e., better fairness-accuracy tradeoff. This is justified by the fact that smaller values of $\epsilon$ increase the certainty of samples with a single value in their prediction set. Specifically for $\epsilon = 0.05$, the Pareto front is closer to our baseline method using the dropout-based uncertainty measure. We observed a similar trend on other datasets and with different base classifiers. These results also support our hypothesis that imposing fairness constraints on the sample with low uncertainty in the sensitive attribute can better improve fairness and accuracy.

On the other hand, using conformal predictions to measure uncertainty, we also evaluated the variant of our hypothesis where the model is trained without fairness constraints using samples with greater uncertainty in the predicted sensitive attributes. This means we trained the model without fairness constraints but using samples whose prediction sets are empty or contain two values. We also trained models without fairness constraints using samples with greater certainty of the sensitive attributes, i.e., samples with single values in their prediction sets. Table 4 shows the fairness and accuracy of Random Forest models trained without fairness constraints using data with different uncertainty levels on the Adult dataset. More specifically, we considered three cases:

- *Certain + Uncertain*: The model is trained using all data points, i.e., the data contain samples with certain and uncertain sensitive attributes.

- *Certain*: The model is trained using only samples with greater certainty of the sensitive attribute. This means the training data contains only samples with a single in the prediction sets.

- *Uncertain*: The model is trained using only samples with greater uncertainty of the sensitive attribute. This means the training data contains only samples whose prediction sets are empty or contain two values.

The results depicted in Table 4 show that, without fairness constraints, the models using samples with greater certainty of sensitive attributes have better accuracy but tend to exacerbate unfairness, while models using data with lower uncertainty significantly improve fairness at the expense of accuracy. More specifically, for $\epsilon = 0.05$, the model using only data with uncertain sensitive attributes achieves fairness in terms of demographic parity of 0.009 compared to 0.12 for the model using data with certain sensitive attributes. This corresponds to a 92% improvement in fairness with a smaller drop of 8% for accuracy. This shows models hardly discriminate against samples with uncertain sensitive attributes, and the fairness-accuracy tradeoffs can be controlled by uncertainty in sensitive attribute space. This also supports our hypothesis under different and reliable uncertainty measures.

## 6 Conclusion

In this work, we introduced FairDSR, a framework to improve the fairness-accuracy tradeoff when only limited demographic information is available. Our method introduces uncertainty awareness in the sensitive attributes classifier. We showed that uncertainty in the attribute classifier plays an important role in the fairness-accuracy tradeoffs achieved in the downstream model with fairness constraints. We showed that enforcing fairness on samples whose sensitive attributes are predicted with low uncertainty can yield models with better fairness-accuracy tradeoffs. Subsequently, we considered different variations around our hypothesis and demonstrated the benefit of uncertainty quantification in the sensitive attribute space for designing fair models under unknown demographic information.

## Impact Statement

The proposed framework shows evidence that models can hardly discriminate against samples with high uncertainty in sensitive attribute prediction. However, our base model relies on the predicted missing sensitive information. Inferring sensitive information can raise ethical concerns and face legal restrictions, especially when individuals do not choose to disclose their sensitive information. The line of methods relying upon proxy attributes or inferring sensitive attributes faces this limitation (Diana et al., 2022; Awasthi et al., 2021; Coston et al., 2019; Liang et al., 2023). For this reason, we proposed a variant of our model (FairDSR (uncertain), Sec. 4.2) that derives a dataset with higher uncertainty in sensitive attributes without using predicted information.In addition, verifying that no group of users is left out when training the model due to the lower uncertainty of their sensitive attributes is important. While we analyzed group representation and showed that groups are still relatively well represented even with a higher uncertainty threshold (Fig. 16 in Supplementary), there are still a risks of disparate impact.

We emphasize that the inference of the sensitive attributes using our proposed method should not be used for any purpose other than bias assessment and mitigation.

### Acknowledgments

SEK is supported by CIFAR and NSERC DG (2021-4086) and UA by NSERC DG (2022-04006). We thank the anonymous reviewers for the feedback, which has helped improve the paper.

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

## Supplementary Material

## A   Limitations

Our method is evaluated mainly on one fairness-enhancing algorithm (i.e., Exponentiated Gradient). It will be interesting to explore if our hypothesis applies to pre-processing and post-processing techniques and with different fairness-enhancing algorithms. Finally, our assumption that the true sensitive attributes are available in the test dataset for fairness evaluation might not be true in several practical scenarios. This might require evaluation using proxy-sensitive attributes. These proxies are likely noisy and might require evaluations using bias assessment methods that effectively quantify fairness violation w.r.t to true sensitive attribute (Chen et al., 2019; Awasthi et al., 2021). On the other hand, it will be interesting to back our empirical findings with theoretical results by characterizing fairness violation bounds based on the uncertainty quantification of the sensitive attribute predictions.

## B   Fairness Metrics & Datasets

In the main paper, we considered the three popular group fairness metrics: demographic parity, equalized odds, and equal opportunity. These metrics aim to equalize different models' performances across different demographic groups. Samples belong to the same demographic group if they share the same demographic information, $A$, e.g., gender and race.

### B.1   Fairness Metrics

**Demographic Parity**   Also known as statistical parity, this measure requires that the positive prediction ($f(X) = 1$) of the model be the same regardless of the demographic group to which the user belongs (Dwork et al., 2012). More formally the classifier $f$ achieves demographic parity if $P(f(X) = 1|A = a) = P(f(X) = 1)$ In other words, the model's outcome should be independent of sensitive attributes. In practice, this metric is measured as follows:

$$\Delta_{\text{DP}}(f) = \left| \underset{x|A=0}{\mathbb{E}}[\mathbb{I}\{f(x) = 1\}] - \underset{x|A=1}{\mathbb{E}}[\mathbb{I}\{f(x) = 1\}] \right| \tag{7}$$

Where $\mathbb{I}(\cdot)$ is the indicator function.

**Equalized Odds**   This metric enforces the True Positive Rate (TPR) and False Positive Rate (FPR) for different demographic groups to be the same $P(f(X) = 1|A = 0, Y = y) = P(f(X) = 1|A = 1, Y = y), :; \forall y \in \{0, 1\}$. The metric is measured as follows:

$$\Delta_{\text{EOD}} = \alpha_0 + \alpha_1 \tag{8}$$

Where,

$$\alpha_j = \left| \underset{x|A=0,Y=j}{\mathbb{E}}[\mathbb{I}\{f(x) = 1\}] - \underset{x|A=1,Y=j}{\mathbb{E}}[\mathbb{I}\{f(x) = 1\}] \right| \tag{9}$$

**Equal Opportunity**   In certain situations, bias in positive outcomes can be more harmful. Therefore, the Equal Opportunity metric enforces the same TPR across demographic (Hardt et al., 2016) and is measured using $\alpha_1$ (Eq. 9).

### B.2   Datasets

In the Adult dataset, the task is to predict if an individual's annual income is greater or less than $50k per year. We also considered the recent version of the Adult dataset (New Adult) for 2018 across different states in US (Ding et al., 2021). For the Compas dataset, the task is to predict whether a defendant will recidivate within two years. The LSAC dataset contains admission records to law school. The task is to predict whether

| Dataset | Size | #features | Domain | Sensitive attribute |
|---------|------|-----------|--------|---------------------|
| Adult Income | 48,842 | 15 | Finance | Gender |
| Compas | 6,000 | 11 | Criminal justice | Race |
| LSAC | 20,798 | 12 | Education | Gender |
| New Adult | 1.6M | 10 | Finance | Gender |
| CelebA | 202,600 | 40 | Image | Gender |

Table 5: Datasets

a candidate will pass the bar exam. The CelebA[5] dataset contains 40 facial attributes of humaine annotated images. We consider the task of predicting *attractiveness* using gender as a sensitive attribute. We randomly sample 20% of the data to train the sensitive attribute classifier ($\mathcal{D}_2$). All the input features are used to train the attribute classifier except for the target variable. Table 5 provides more details about each dataset and sensitive attributes used.

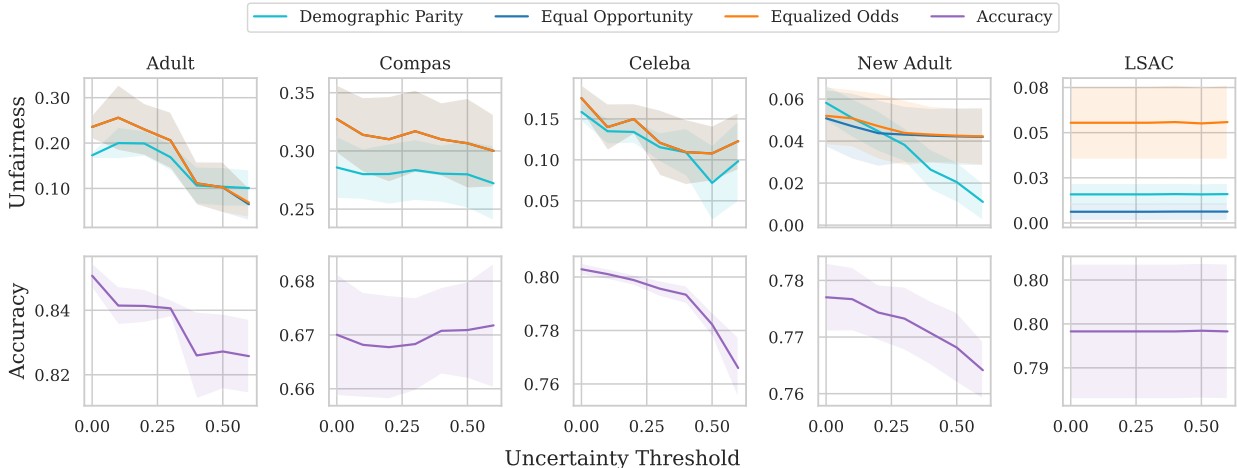

Figure 8: Training Logistic Regression without fairness constraints using samples with high uncertainty of sensitive attribute predictions. For each uncertainty threshold $H$, the model is trained on samples with uncertainty $\geq H$. The training is done seven times, and the average fairness (first row) and accuracy (second row) are reported. Shaded represents the standard deviation.

## C Additional Results

### C.1 Impact of the group-labeled ratio.

In all the previous experiments, we considered the ratio of the group-labeled dataset ($\mathcal{D}_2$) as 20% of the original dataset. In many real-world scenarios, data labeling is costly, especially given the rising concerns surrounding privacy. As these concerns intensify, fewer users may be inclined to disclose their sensitive attributes. It is, therefore, important to study the impact of the group-labeled ratio on the performance of our method. In this ablation study, we consider the attribute classifier trained with different ratios of $\mathcal{D}_2$, i.e., 3%, 5%, 10%, 15%, and 20% of the original dataset. We experiment on the Adult, wherein 3% represents around 1,465 data points among the 48,842 data points available.

---

[5]http://mmlab.ie.cuhk.edu.hk/projects/CelebA.html

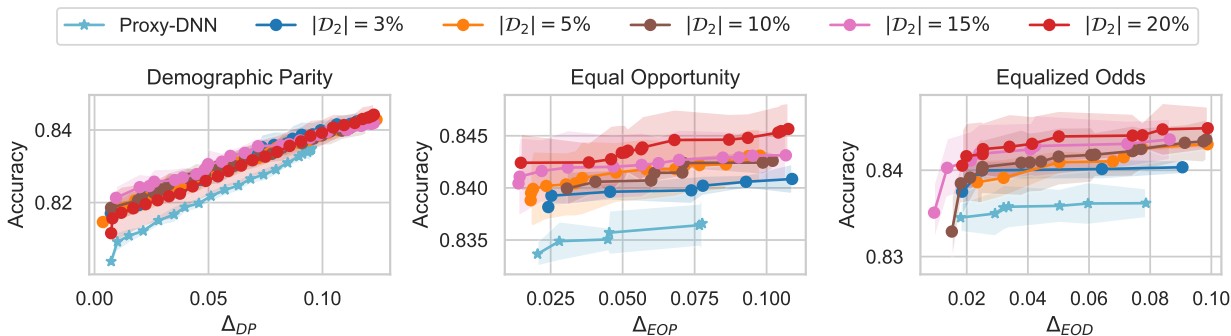

Figure 9: **Study of group-labeled ratio**. The sensitive attribute classifier is trained using different ratios of the dataset with sensitive attributes ($\mathcal{D}_2$).

Figure 9 shows the Pareto front for different group-labeled ratios. Our findings remain consistent even in the most demographic scare regime. While we observe a slight drop in accuracy for smaller group-labeled ratios, the fairness performances are maintained. The Pareto fronts still dominate the model directly using the predicted sensitive attributes. These results suggest that our hypothesis holds even when very limited demographic information is available, as long as the dataset is predictive for sensitive attributes. On the other hand, even if the dataset is not predictive enough for the sensitive attributes, our results show that training a model without fairness constraints will exhibit less disparities across the unknown demographic groups.

## C.2 Results on Nonlinear Classifier.

Table 6 compares with other baselines on the CelebA dataset with logistic regression as the base classifier. In the main paper, we compared existing methods using Logistic Regression as the base classifier. We performed experiments with a more complex non-linear model to analyze its impact on the performance of different methods. We considered a Multi-Layer Perceptron (MLP) with one hidden layer of 32 units and Relu as the activation function for all the baselines. On the Adult dataset, Table 8 shows that when using a more complex model, our method (FairDSR (certain)) still provides Pareto dominants points in terms of fairness and accuracy compared to other baselines. At the same time, we observed an improvement in the accuracy of other methods due to the increased model capacity.

| Method | Accuracy | $\Delta_{DP}$ | $\Delta_{EOP}$ | $\Delta_{EOD}$ |
|---|---|---|---|---|
| Vanilla (without fairness) | $0.803 \pm 0.002$ | $0.176 \pm 0.010$ | $0.183 \pm 0.015$ | $0.183 \pm 0.015$ |
| Vanilla (with fairness) | $0.782 \pm 0.001$ | $0.008 \pm 0.005$ | $0.018 \pm 0.014$ | $0.017 \pm 0.014$ |
| Proxy-DNN | $0.800 \pm 0.001$ | $0.008 \pm 0.036$ | $0.011 \pm 0.022$ | $0.036 \pm 0.011$ |
| FairDA | $0.802 \pm 0.002$ | $0.155 \pm 0.010$ | $0.165 \pm 0.018$ | $0.165 \pm 0.018$ |
| CGL | $0.782 \pm 0.002$ | $0.008 \pm 0.005$ | $0.019 \pm 0.011$ | $0.020 \pm 0.009$ |
| ARL | $\mathbf{0.803} \pm 0.002$ | $0.157 \pm 0.010$ | $0.157 \pm 0.010$ | $0.166 \pm 0.016$ |
| CVarDRO | $0.781 \pm 0.002$ | $0.155 \pm 0.010$ | $0.162 \pm 0.016$ | $0.162 \pm 0.016$ |
| KSMOTE | $0.773 \pm 0.008$ | $0.020 \pm 0.067$ | $0.110 \pm 0.082$ | $0.144 \pm 0.068$ |
| DRO | $0.796 \pm 0.006$ | $0.142 \pm 0.020$ | $0.152 \pm 0.020$ | $0.129 \pm 0.028$ |
| FairDSR (weighted) | $0.799 \pm 0.004$ | $0.038 \pm 0.010$ | $0.020 \pm 0.012$ | $0.033 \pm 0.013$ |
| FairDSR (uncertain) | $0.782 \pm 0.004$ | $0.071 \pm 0.046$ | $0.107 \pm 0.033$ | $0.107 \pm 0.033$ |
| FairDSR (certain) | $0.793 \pm 0.000$ | $\mathbf{0.003} \pm 0.000$ | $\mathbf{0.001} \pm 0.001$ | $\mathbf{0.007} \pm 0.002$ |

Table 6: Results on the CelebA dataset.

| Method | Accuracy | $\Delta_{\mathrm{DP}}$ | $\Delta_{\mathrm{EOP}}$ | $\Delta_{\mathrm{EOD}}$ |
|---|---|---|---|---|
| Vanilla (without fairness) | $0.793 \pm 0.007$ | $0.014 \pm 0.005$ | $0.005 \pm 0.005$ | $0.049 \pm 0.026$ |
| Vanilla (with fairness) | $0.796 \pm 0.009$ | $0.004 \pm 0.004$ | $0.002 \pm 0.001$ | $0.025 \pm 0.016$ |
| Proxy-DNN | $0.792 \pm 0.009$ | $0.018 \pm 0.009$ | $0.004 \pm 0.001$ | $0.040 \pm 0.005$ |
| FairRF | $0.753 \pm 0.120$ | $0.021 \pm 0.013$ | $0.016 \pm 0.017$ | $0.044 \pm 0.015$ |
| FairDA | $0.716 \pm 0.210$ | $\mathbf{0.001} \pm 0.000$ | $0.000 \pm 0.005$ | $0.003 \pm 0.004$ |
| CGL | $0.765 \pm 0.022$ | $0.005 \pm 0.004$ | $0.003 \pm 0.001$ | $0.035 \pm 0.017$ |
| ARL | $\mathbf{0.807} \pm 0.024$ | $0.014 \pm 0.015$ | $0.009 \pm 0.014$ | $0.037 \pm 0.022$ |
| CVarDRO | $0.776 \pm 0.052$ | $0.024 \pm 0.010$ | $0.019 \pm 0.014$ | $0.045 \pm 0.015$ |
| KSMOTE | $0.655 \pm 0.055$ | $0.022 \pm 0.034$ | $0.030 \pm 0.022$ | $0.060 \pm 0.018$ |
| DRO | $0.580 \pm 0.220$ | $0.023 \pm 0.014$ | $0.021 \pm 0.017$ | $0.038 \pm 0.020$ |
| FairDSR (weighted) | $0.806 \pm 0.001$ | $0.008 \pm 0.005$ | $0.004 \pm 0.002$ | $0.035 \pm 0.018$ |
| FairDSR (uncertain) | $0.794 \pm 0.001$ | $0.015 \pm 0.002$ | $0.006 \pm 0.001$ | $0.055 \pm 0.000$ |
| FairDSR (certain) | $0.805 \pm 0.001$ | $\mathbf{0.001} \pm 0.002$ | $\mathbf{0.000} \pm 0.001$ | $\mathbf{0.002} \pm 0.000$ |

Table 7: Results on the LSAC dataset.

| Method | Accuracy | $\Delta_{\mathrm{DP}}$ | $\Delta_{\mathrm{EOP}}$ | $\Delta_{\mathrm{EOD}}$ |
|---|---|---|---|---|
| Vanilla (without fairness) | $0.853 \pm 0.004$ | $0.183 \pm 0.019$ | $0.100 \pm 0.025$ | $0.102 \pm 0.023$ |
| Vanilla (with fairness) | $0.801 \pm 0.009$ | $0.006 \pm 0.004$ | $0.049 \pm 0.011$ | $0.017 \pm 0.007$ |
| Proxy-DNN | $0.795 \pm 0.009$ | $0.029 \pm 0.004$ | $0.067 \pm 0.020$ | $0.042 \pm 0.011$ |
| FairRF | $\mathbf{0.853} \pm 0.002$ | $0.164 \pm 0.009$ | $0.077 \pm 0.026$ | $0.091 \pm 0.013$ |
| FairDA | $0.813 \pm 0.014$ | $0.118 \pm 0.023$ | $0.091 \pm 0.050$ | $0.099 \pm 0.037$ |
| CGL | $0.800 \pm 0.014$ | $0.009 \pm 0.002$ | $0.017 \pm 0.021$ | $0.032 \pm 0.010$ |
| ARL | $0.851 \pm 0.003$ | $0.166 \pm 0.015$ | $0.087 \pm 0.019$ | $0.090 \pm 0.016$ |
| CVarDRO | $0.850 \pm 0.003$ | $0.183 \pm 0.018$ | $0.095 \pm 0.027$ | $0.101 \pm 0.026$ |
| KSMOTE | $0.814 \pm 0.020$ | $0.201 \pm 0.055$ | $0.120 \pm 0.021$ | $0.130 \pm 0.023$ |
| DRO | $0.837 \pm 0.016$ | $0.232 \pm 0.057$ | $0.110 \pm 0.057$ | $0.140 \pm 0.045$ |
| FairDSR (weighted) | $0.849 \pm 0.027$ | $0.018 \pm 0.010$ | $0.061 \pm 0.034$ | $0.047 \pm 0.032$ |
| FairDSR (uncertain) | $0.801 \pm 0.027$ | $0.110 \pm 0.022$ | $0.067 \pm 0.027$ | $0.059 \pm 0.024$ |
| FairDSR (certain) | $0.818 \pm 0.004$ | $\mathbf{0.009} \pm 0.008$ | $\mathbf{0.028} \pm 0.020$ | $\mathbf{0.027} \pm 0.017$ |

Table 8: Results on the Adult dataset using an MLP with a hidden layer of 34 units as base classifier.

### C.3 Comparison with Other Baselines

To assess the effect of the attribute classifier over the performances of downstream classifiers with fairness constraints w.r.t the proxy, we also performed extensive comparisons with different methods of obtaining the missing sensitive attributes:

- **Ground truth sensitive attribute**. We considered the case where the sensitive attribute is fully available and trained the model with fairness constraints w.r.t the ground truth. This represents the ideal situation where all the assumptions about the availability of demographic information are satisfied. This baseline is expected to achieve the best trade-offs.

- **Proxy-KNN**. Here the missing sensitive attributes are handled by data imputation using the k-nearest neighbors (KNN) of samples with missing sensitive attributes.

- **Proxy-DNN**. For this baseline, an MLP is trained on $\mathcal{D}_2$ to predict the sensitive attributes without uncertainty awareness. The network architecture and the hyperparameter are the same as for the student in our model.

- CGL (Jung et al., 2022): Similar to our setup, this method also uses an attribute classifier and replaces uncertain predictions with random sampling from the empirical conditional distribution of the sensitive attributes given the non-sensitive attributes $P(A|X,Y)$.

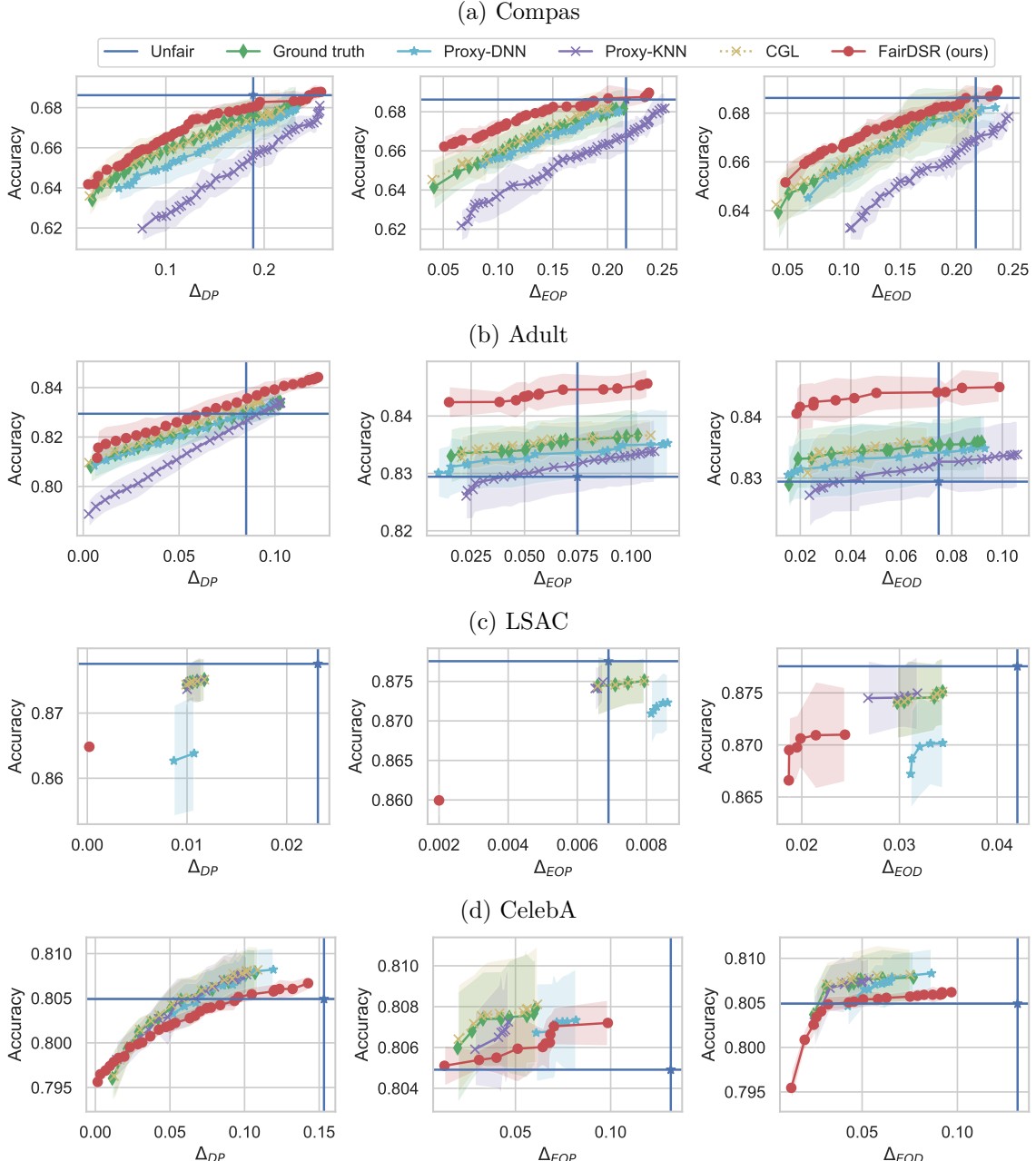

Figure 10: Accuracy-fairness trade-offs for various fairness metrics ($\Delta_{\mathrm{DP}}$, $\Delta_{\mathrm{EOP}}$, $\Delta_{\mathrm{EOD}}$) and proxy sensitive attributes. The top-left is the best (highest accuracy with the lowest unfairness). Curves are created by sweeping a range of fairness coefficients $\lambda$, taking the median of 7 runs per $\lambda$, and computing the Pareto front. The exponentiated gradient is the fairness mechanism with Random Forests as the base classifier. The standard deviations are shaded in the figures.

For fairness-enhancing mechanisms, we considered the Fairlean (Bird et al., 2020) implementation of the exponentiated gradient (Agarwal et al., 2018) and adversarial debiasing (Zhang et al., 2018) (Section 3). We

used various base classifiers for the exponentiated gradient, including logistic regression, random forest, and gradient-boosted trees. Random forest was initialized with a maximum depth of 5 and minimum sample leaf of 10, and default parameters were used for logistic regression without hyperparameter tuning. The same models and hyperparameters were used across all the datasets. Adversarial debiasing works for demographic parity and equalized odds. The architecture of the classifier, the adversary, and other hyperparameters used is the same as recommended by the original paper (Zhang et al., 2018). We evaluate the fairness-accuracy trade-off of every baseline by analyzing the accuracy achieved in different fairness regimes, i.e., by varying the parameter $\epsilon \in [0, 1]$, controlling the balance between fairness and accuracy. For a value of $\epsilon$ close to 0, the label classifier is enforced to achieve higher accuracy, while for a value close to 1, it is encouraged to achieve lower unfairness. For each value of $\epsilon$, we trained each baseline 7 times on a random subset of $\mathcal{D}_1$ (70%) using their predicted sensitive attributes, and the accuracy and fairness are measured on the remaining subset (30%), where we assumed that the joint distribution $(X, Y, A)$ is available for fairness evaluation. The results are averaged, and the Pareto front is computed.

Figure 10 shows the Pareto front of the exponentiated gradient method on the Adult, Compas, and LSAC datasets using Random Forests as the base classifier. The figure shows the trade-off between fairness and accuracy for the different methods of inferring the missing sensitive attributes. From the results, we observe on all datasets and across all fairness metrics that data imputation can be an effective strategy for handling missing sensitive attributes, i.e., this fairness mechanism can efficiently improve the model's fairness with respect to the true sensitive attributes, although fairness constraints were enforced on proxy-sensitive attributes. However, we observe a difference in the fairness-accuracy trade-off for each attribute classifier. The KNN-based attribute classifier has the worst fairness-accuracy trade-off on all datasets and fairness metrics. This shows that assigning sensitive attributes based on the nearest neighbors does not produce sensitive attributes useful for achieving a trade-off close to the ground truth. While the DNN-based attribute classifier produces a better trade-off, it is still suboptimal compared to the ground truth-sensitive attributes. We observed similar results with different baseline models, such as logistic regression and gradient-booted trees, and for adversarial debiasing as the fairness mechanism. In contrast, our method consistently achieves a better trade-off on all datasets and the fairness metrics considered. Similar results are obtained on the exponentiated gradient with logistic regression and gradient-boosted trees as base classifiers and adversarial debiasing (see Section 6). The uncertainty threshold's choice depends on the dataset's bias level, i.e., the level of information about the sensitive attribute encoded in the feature space.

Figures 11(d), 10(d), and 12(d), depict the Pareto front of various baselines on the CelebA dataset. It shows that models trained with imputed sensitive attributes via KNN consistently achieve comparable tradeoffs to models trained with fairness constraints based on the true sensitive attribute. This could be explained by the fact that gender clusters are perfectly defined in the latent space. We observed that KNN-based imputation achieved 95% accuracy in predicting gender. Conversely, the figures illustrate that our method outperforms baselines using ground truth-sensitive attributes and imputation methods, yielding more Pareto-dominant points. This highlights the advantages of applying fairness constraints to samples with low uncertainty in the sensitive attributes. Furthermore, Figure 14 (c) shows decreasing the uncertainty threshold further improves fairness while preserving the accuracy. We note the CelebA dataset can raise ethical concerns and is used only for evaluation purposes. For instance, predicting the attractiveness of a photo using other facial attributes as sensitive attributes can still harm individuals even if the model predicting attractiveness is not *biased*.

**Exponentiated gradient with different baseline classifiers**  Figure 11 and 12 show fairness-accuracy trade-offs achieved by the exponentiated gradient with logistic regression and gradient-boosted trees, respectively. Similar to the results presented in the main paper, our method achieves better fairness-accuracy trade-offs.

Figure 14 shows the accuracy-fairness trade-off Exponentiated gradient using gradient-boosted trees as the base classifier for various uncertainty thresholds, the true sensitive attributes, and the predicted sensitive attributes with DNN. The results obtained are similar to random forests as the base classifier. The smaller uncertainty threshold produced the best trade-off in a high-bias regime like the Adult dataset. On datasets that do not encode much information about the sensitive attributes (most samples have high uncertainty), such as the New Adult and LSAC datasets, the accuracy decreases as the uncertainty threshold reduces

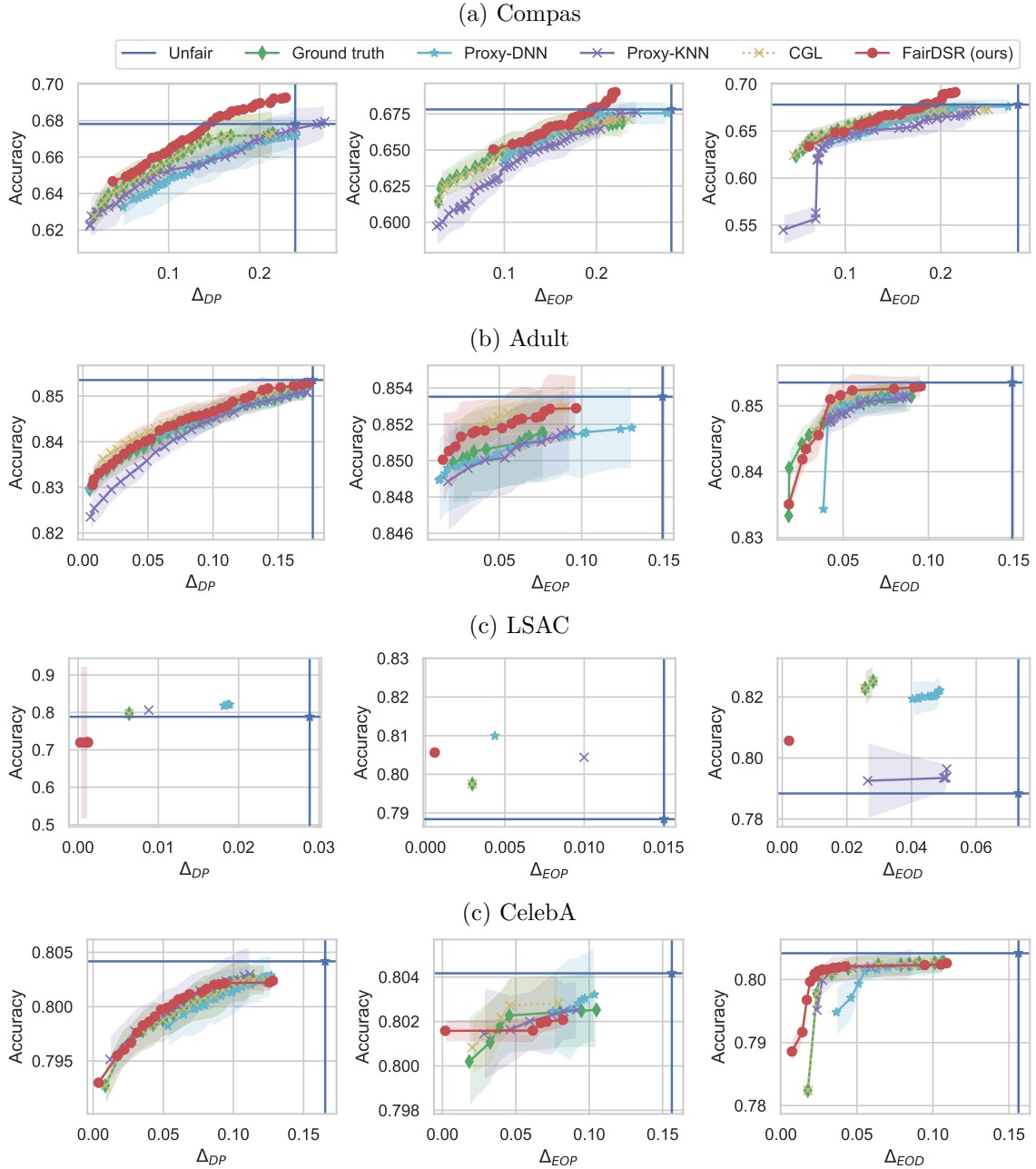

Figure 11: Accuracy-fairness trade-offs for various fairness metrics ($\Delta_{\mathrm{DP}}$, $\Delta_{\mathrm{EOP}}$, $\Delta_{\mathrm{EOD}}$) and proxy-sensitive attributes. Top-left is the best (Highest accuracy with the lowest unfairness). The fairness mechanism is the Exponentiated gradient with logistic regression as the base classifier on the Compas (a), Adult (b), LSAC (c), and CelebA (c)datasets. The standard deviation is shaded in the figure.

while fairness is improved or maintained. On the LSAC dataset (Figure 14(d)), we observe that increasing the uncertainty threshold results in a much higher drop in accuracy. This is explained by the high average uncertainty (0.66), and using a smaller threshold removes most of the data.

**Experiments with adversarial debiasing** Figure 15 shows the trade-offs for adversarial debiasing. Our methods achieve a better trade-off on the Adult datasets, while for the Compas dataset, the ground-truth sensitive achieves a better trade-off. It is worth noting that adversarial debiasing is unstable to train.

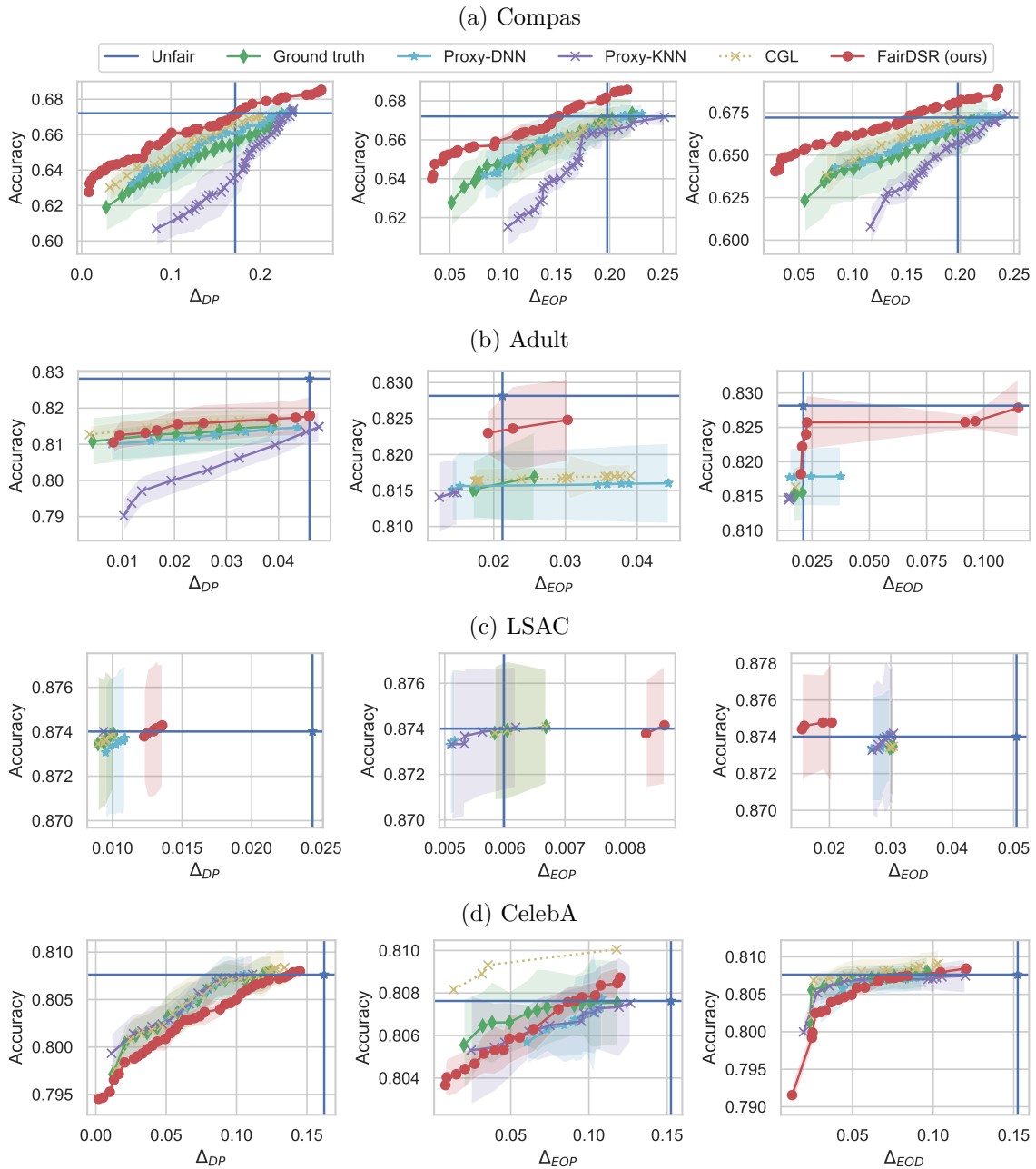

Figure 12: Accuracy-fairness tradeoffs for various fairness metrics ($\Delta_{\mathrm{DP}}$, $\Delta_{\mathrm{EOP}}$, $\Delta_{\mathrm{EOD}}$) and proxy sensitive attributes. The fairness mechanism used is the Exponentiated gradient with gradient-boosted trees as the base classifier on the Compas (a) Adult (b), (c) LSAC, and (d) CelebA datasets. The standard deviation is shaded in the figure.

# D    Uncertainty Estimation of Different Demographic Groups

In the main paper, we showed that when the dataset does not encode enough information about the sensitive attributes, the attribute classifier has, on average, greater uncertainty in the predictions of sensitive attributes. This encourages a choice of a higher uncertainty threshold to keep enough samples to maintain the accuracy, i.e., to prune out only the most uncertain samples. Figure 16 shows that the gap between demographic groups can increase as a smaller uncertainty threshold is used. This is explained by the fact that the model is more

confident about samples from well-represented groups than samples from under-represented groups. While this gap between demographic groups can increase, our results show there are still enough samples from the disadvantaged group with reliable sensitive attributes. Thus, tuning the uncertainty threshold can result in a model that achieves a better trade-off between accuracy and various fairness metrics. Note that we observed the same trend for the LSAC dataset. The average uncertainty is 0.66, and the minimum uncertainty is 0.62. We also observed that group representation remains consistent (35% difference) when using the average uncertainty value.

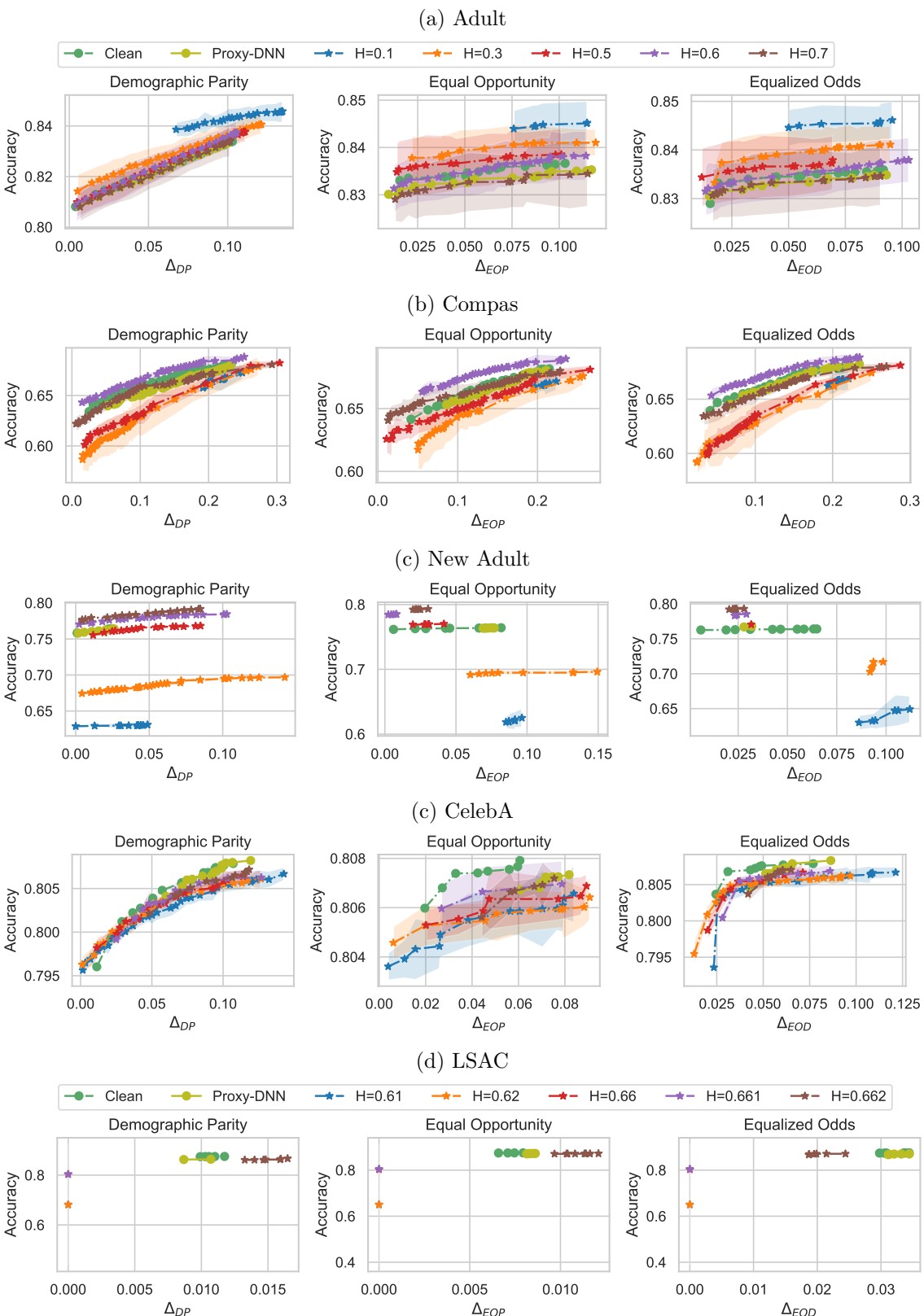

Figure 13: The impact of the uncertainty threshold $H$ on the fairness-accuracy trade-off. For the exponentiated gradient with Random Forest as the base classifier for the (a) Adult, (b) Compas, (c) New Adult, (c) CelebA dataset, and (d) LSAC datasets.

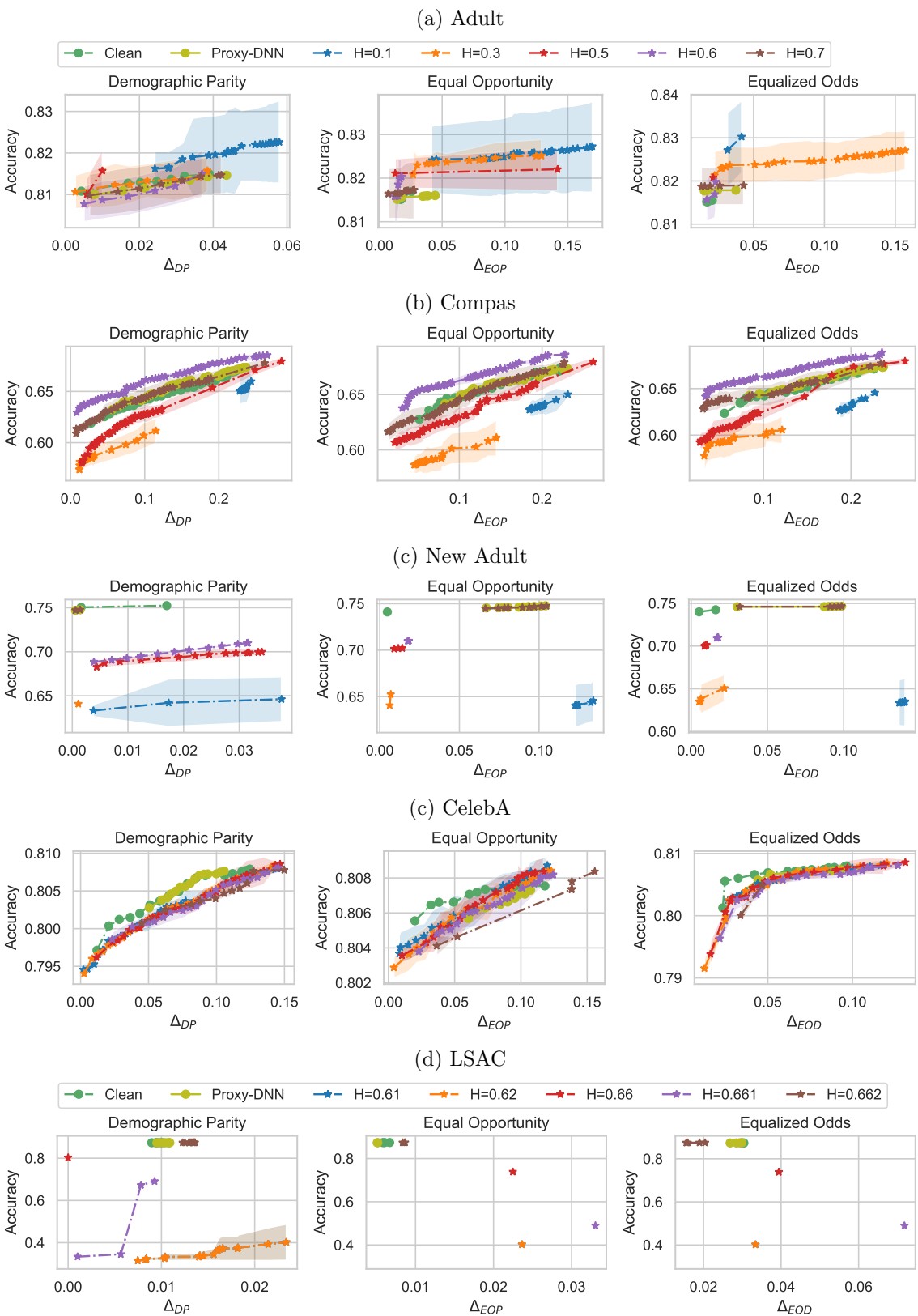

Figure 14: Exponentiated gradient with gradient-boosted trees as the base classifier. The impact of the uncertainty threshold $H$ on the fairness-accuracy trade-off for the (a) Adult, (b) Compas, (c) New Adult, (c) CelebA dataset, and (d) LSAC datasets.

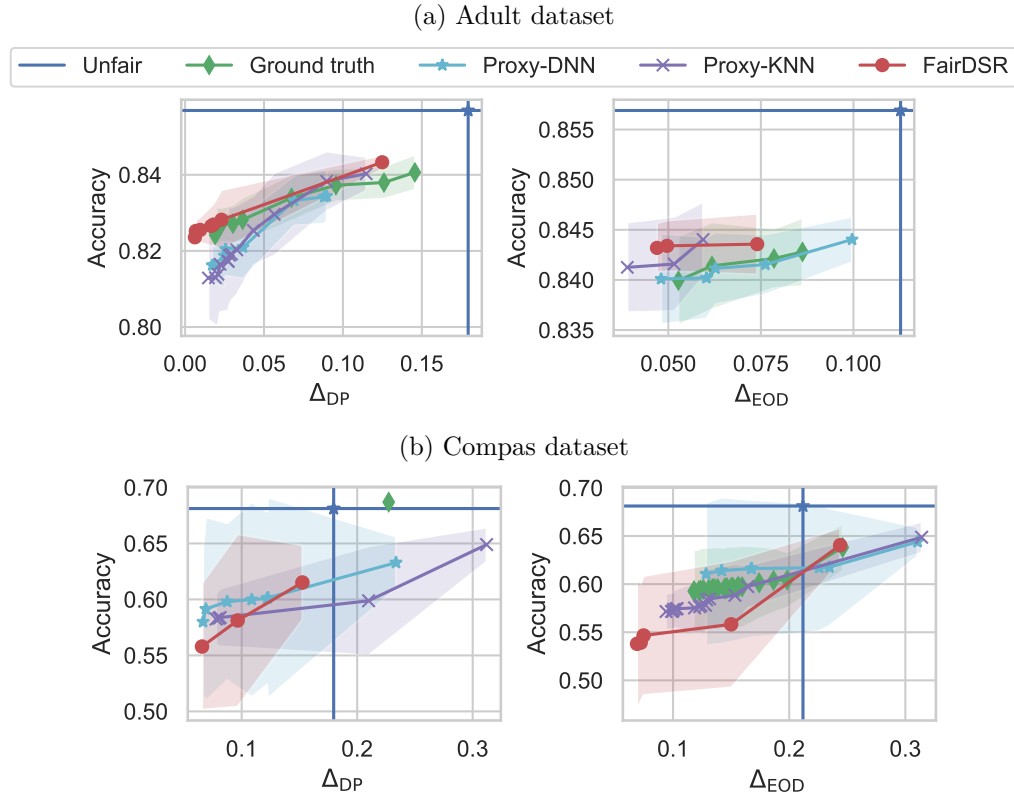

Figure 15: Adversarial debiasing. Accuracy-fairness trade-offs for various fairness metrics ($\Delta_{\text{DP}}$, $\Delta_{\text{EOP}}$) and proxy-sensitive attributes.

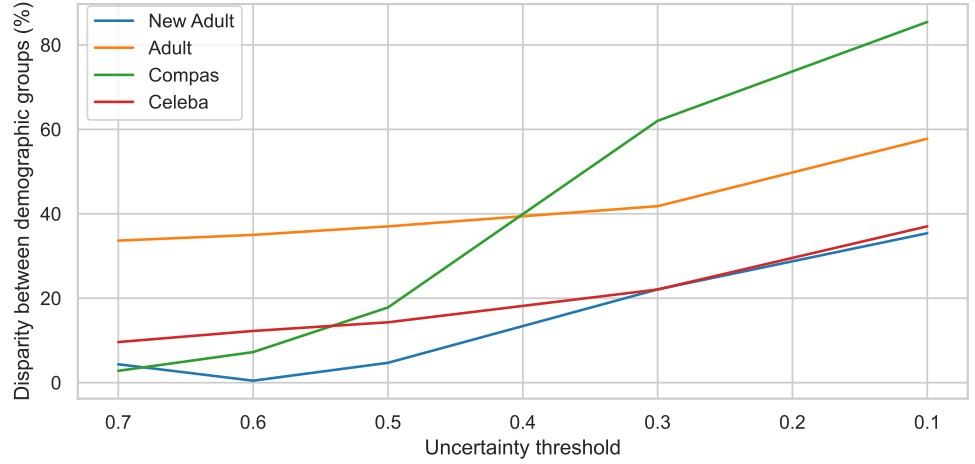

Figure 16: Demographic group representation in each dataset for different uncertainty thresholds. The gap between groups increases as the threshold becomes smaller. The plot reveals there are samples from the minority group that exhibit lower uncertainty in the prediction of their sensitive attributes.

