# OpenReview forum: "Fairness Under Demographic Scarce Regime"
_TMLR — Accepted by TMLR_

### Review · Reviewer_6s5W · 2024-06-25

**Summary Of Contributions:**

The paper considers the problem of training a classifier f : R^d -> {0,1} under some fairness constraint from R^d to some other binary demographic attribute (they consider equalized odds, equal opportunity, and demographic parity).
The starting point is the observation that when ability to predict the demographic attribute has a lot of uncertainty, then optimizing for the main classification tasks induces less unfairness.
So in the context of training main classifier f, subject to some fairness constraint, the paper introduces the following idea:
  1. assess uncertainty on each data points wrt how well it can predict the demographic attribute
  2. only train f wrt fairness for data x with uncertainty below some threshold.

The paper sets of an experimental design with careful test train split, finds several data sets where this task is relevant, provides multiple baseline methods from the past few years, and ultimately shows that their approach often empirically performs the best.

The experiments appear well organized and properly done, the writing is (mostly) clear, and results are convincing.

**Audience:**

Yes

**Broader Impact Concerns:**

Some data points or attributes that have weak association with demographics may be ignored because of this uncertainty.  This could systematic, and a as a result a group could be left out of classification because of this.  I think this effect is probably neutral in the grand scheme of things, but perhaps should be discussed more.

**Claims And Evidence:**

Yes

**Requested Changes:**

1.  Some of the related work discussion could be improved to not just summarize the main idea in papers, but to also explain why these ideas do not need to be directly compared against in more depth.  For instance, in page 3, the Diana etal (2020) paper is described as
  "...training a model to predict the sensitive attributes can be a good substitute for the ground truth sensitive attributes when the latter is missing"
 It would be useful to explain in the context of that paper's discussion why we cannot simply apply their ideas.  And similar for other papers in that paragraph.

2.  It was not clear in the student model on page 5 if the threshold R was "predefined" and fixed (as said above equation 1), or "updated" through training or cross-validation (as implied in the last line of the page, after equation 3).
  Moreover, explain how R relates to H in equation (5) and discussed later in the paper.

**Strengths And Weaknesses:**

Strengths:
 - The observation is nice.
 - The writing is clear enough.
 - The experiments a convincing that the core idea works

Weaknesses:
 - There are a few design choices which could have been done differently (choice of classifier, notion of fairness).  But this is a weak weakness, as all choices are reasonable.

---

> ### Author Response · Authors · 2024-07-13
> **Response to Reviewer 6s5W**
>
> We thank the reviewer for the detailed feedback and comments, which helped improve the clarity of our work. The requested changes are highlighted in blue in the revised manuscript and outlined below.
>
> > Some of the related work discussion could be improved to not just summarize the main idea in papers, but to also explain why these ideas do not need to be directly compared against in more depth [...]
>
>
> We updated the related work section and clarified the position of our work w.r.t to each related work.
>
> > It was not clear in the student model on page 5 if the threshold R was "predefined" and fixed (as said above equation 1), or "updated" through training or cross-validation (as implied in the last line of the page, after equation 3).
> Moreover, explain how R relates to H in equation (5) and discussed later in the paper.
>
>
> Sorry for the confusion. We have clarified that the threshold R was not fixed during the training. Moreover, we emphasized that the threshold $R$ in the first step is only used to train the attributes classifier while ensuring the student and teacher remain consistent for samples with uncertainty lower than $R$. The threshold $H$ in the second step is used at test time to select samples with low uncertainty, and its value can be tuned over a validation set.
>
> > Some data points or attributes that have weak association with demographics may be ignored because of this uncertainty. This could systematic, and a as a result a group could be left out of classification because of this. I think this effect is probably neutral in the grand scheme of things, but perhaps should be discussed more.
>
> We have acknowledged this risk in the broader impact dispatch discussion. Furthermore, we have analyzed the group representation in the data with greater uncertainty for different uncertainty thresholds and observed that all groups are still relatively well represented even in the higher uncertainty threshold (Fig. 16 in the supplementary).
>
> Again, we thank the reviewer for the feedback and we will be happy to address any remaining concerns.

---

> > ### Comment · Reviewer_6s5W · 2024-07-13
> >
> > Thanks for carefully addressing my comments.  I am satisfied, and I think the paper is stronger after these changes.

---

### Review · Reviewer_JCP5 · 2024-07-01

**Summary Of Contributions:**

The main method presented incorporates uncertainty quantification in a sensitive-attribute classifier to achieve a better fairness-accuracy tradeoff. The key idea behind this method is that samples with lower uncertainty should be used for fairness constraints. Uncertainty quantification is achieved through Monte Carlo dropout. Empirical results support the hypothesis that enforcing fairness constraints with certain data gives more desirable results compared to enforcing these constraints using uncertain or all data. Although the uncertainty quantification mechanism used is somewhat straightforward, the finding that uncertainty quantification is useful in this setting is a valuable contribution.

**Audience:**

Yes

**Broader Impact Concerns:**

The impact statement clearly states that this work should only be used for bias assessment and mitigation. This section also mention that this method may face legal restrictions.

**Claims And Evidence:**

Yes

**Requested Changes:**

The confidence interval paragraph (at the end of Section 5) could use additional details. What is the form of the interval/how is it being calculated? This information should be included. I believe that a prediction interval would be more appropriate in this context (using Conformal Prediction, for example). My instinct is that this would quantify uncertainty that is not captured with the current methods. Including such a comparison would strengthen the work, while some discussion on why a prediction interval would not be necessary seems critical.

Minor Typographical Errors and Writing Suggestions:
-	Second paragraph on page 2 mentions “In the first phase” but then does not mention a second phase. I had to reread to make sure I didn’t skip over text about the second phase. This could be combined with the text 2 paragraphs later, giving a full overview of the framework. The paragraph in the middle is heavy on details of the first phase which are not necessary to go into before a full overview.
-	The final sentence of Section 1 seems to be missing an “and”.
-	Bottom of page 4 the Gal and Ghahramani in-line citation should be in parentheses

**Strengths And Weaknesses:**

Strengths:
-	This paper clearly states a main hypothesis and answers this hypothesis with supporting results. In general, this paper is well written and easy to follow. There are some minor typos and suggestions I mention in Requested Changes. The takeaway message of this work is clear.
-	Although I am not entirely familiar with all baselines presented (nor would I be able to point to any other baselines that should be considered), the empirical results section contains comparisons to many other existing methods. The ablation study suggests that the demonstrated gain in performance is not due to some hyperparameter-hacking and makes the presented results trustworthy. Despite not going through the provided code extensively, I am confident in the reproducibility of the results in this paper.
-	I have no concerns regarding correctness, as no novel theory is proposed.
Weaknesses:
-	The technical novelty is rather limited. The framework combines a semi-supervised learning technique with MC dropout. Technical novelty is limited to applying this combination to a task that previously had not been considered. The uncertainty that the technique captures is geared towards capturing uncertainty in the model fit, rather than of the prediction (i.e. if the sensitive attribute has high levels of variability itself, I’m not convinced that MC Dropout will capture that). I could not identify any theoretical insights to support the main claim and have to rely on the empirical results alone.

---

> ### Author Response · Authors · 2024-07-13
> **Response to Reviewer JCP5**
>
> We thank the reviewer for the careful assessment of our work and the thorough feedback. The requested changes are highlighted in blue in the revised manuscript and outlined below.
>
>
> > The confidence interval paragraph (at the end of Section 5) could use additional details. What is the form of the interval/how is it being calculated?
>
>
> We construct the confidence interval using a threshold $\tau \in [0.5, 1]$ over the predicted probability $P(\hat{A} = a|X =x)$, such that samples with higher uncertainty have their prediction probability closer to $0.5$. A given data point $x$ is in the high uncertainty set if  $ 1-\tau < P(\hat{A} = a|X =x) < \tau$, while $x$ is in the low uncertainty set otherwise. More specifically, in this experiment, we derive the subset of data points with low uncertainty of the sensitive attribute prediction as follows :
>
> $$
>      (x, y, f(x)) \in \mathcal{D}_1' \operatorname{if} \;
>              P(\hat{A} = a|X =x) \in [0, 1- \tau], \text{or} P(\hat{A} = a|X =x) \in [\tau, 1]
> $$
>
> We have provided the details in the revised manuscript.
>
>
> > I believe that a prediction interval would be more appropriate in this context (using Conformal Prediction, for example). My instinct is that this would quantify uncertainty that is not captured with the current methods. Including such a comparison would strengthen the work, while some discussion on why a prediction interval would not be necessary seems critical.
>
>
> Thank you for the suggestion. Conformal predictions guarantee that the true label is within the prediction set with a probability of $1-\epsilon$. However, while the size of the prediction set is an indicator of the model uncertainty, it does not provide enough information, especially for binary classification [1].
> We provided an ablation study on conformal prediction as the uncertainty measure. In particular, we constructed prediction sets of sensitive attributes and considered samples with low uncertainty as samples whose prediction sets contain single values for different coverage $\epsilon \in \{0.05, 0.1, 0.2\}$. Figure 7 of the revised manuscript shows that fair models with smaller values of $\epsilon$ (i.e., using samples with lower certainty) result in better Pareto points. Moreover, for $\epsilon=0.05$, the Pareto front is closer to our baseline method using dropout-based uncertainty measure, while our baseline method provides better results than conformal prediction. This is explained by our training process of the attribute classifier with uncertainty awareness that can provide more consistent uncertainty measures using consistency loss. Moreover, MC Dropout also captures the variability in predicted sensitive attributes since the entropy is computed over the output of an ensemble of subnetworks. Thus, samples receiving conflicting predictions across subnetworks indicate higher uncertainty in the predicted attributes, which is captured by the entropy over the outputs of the subnetworks’.
>
> On the other hand, we evaluated other aspects of our hypothesis using conformal prediction as an uncertainty measure. Specifically, we studied the fairness and accuracy performance of a model trained without fairness constraints but using data with different levels of uncertainty. The results in Table 4 show that the models using samples with greater certainty (prediction sets containing a single value) of sensitive attributes tend to exacerbate unfairness, while models using data with lower uncertainty provide better fairness results. This shows models hardly discriminate against samples with uncertain sensitive attributes, supporting our hypothesis under different and reliable uncertainty measures such as conformal prediction.
>
> We have provided more details in the revised manuscript on pages 14-15.
>
> > Minor Typographical Errors and Writing Suggestions
>
>
> We have fixed the inconsistency in the description of each phase of the method and provided a full overview before giving the details of each phase.
>
> We thank the reviewer for the feedback, which has helped improve the manuscript, and we will be happy to address any remaining concerns.
>
> References:
>
> [1] Gupta, C., Podkopaev, A., & Ramdas, A. (2020). Distribution-free binary classification: prediction sets, confidence intervals and calibration. Advances in Neural Information Processing Systems, 33, 3711-3723.

---

> > ### Comment · Reviewer_JCP5 · 2024-07-24
> > **Response to Author Response**
> >
> > Thank you for your detailed response and revisions. The additional details for the confidence intervals in the revised manuscript are clear and make the presentation stronger. Indeed, prediction set sizes from conformal prediction for binary classification do not add much information, thank you for bringing this point to my attention. The addition of conformal prediction in simulation studies reinforces the main hypothesis, and the presentation of these results is well written. I have no further questions or concerns from the author(s).

---

### Review · Reviewer_sdai · 2024-07-05

**Summary Of Contributions:**

This paper proposes a novel framework called FairDSR to enhance fairness in machine learning models when demographic information is incomplete or unavailable. The key contributions include:

-	Uncertainty-Aware Attribute Classifier: An approach using self-ensembling and Monte Carlo dropout to reliably predict demographic attributes with quantified uncertainty.
-	Fairness Constraints: Enforcing fairness constraints only on samples with low uncertainty in their predicted demographic attributes.
-	Demonstrating the framework's effectiveness on five datasets, outperforming traditional methods.
-	Exploring different methods such as weighted fairness constraints and training on high-uncertainty samples without fairness constraints.

**Audience:**

Yes

**Broader Impact Concerns:**

The authors have already discussed the ethical impact of the paper.

**Claims And Evidence:**

Yes

**Requested Changes:**

Please refer to the weaknesses mentioned above.

**Strengths And Weaknesses:**

Strengths:
- This paper proposes an intuitive and interesting approach to handle fairness where demographic data is incomplete or unavailable.

- The use of uncertainty-aware models and self-ensembling techniques is well-founded and effectively improves the reliability of demographic attribute predictions. The validation on multiple datasets and ablation studies strengthens the effectiveness of the proposed method.

- The exploration of different variants of the method, such as the weighted approach and training without fairness constraints is interesting.


Weaknesses:
- The effectiveness of the method heavily relies on the accuracy and reliability of the uncertainty measure. The performance of the framework might degrade if these measure are not well-calibrated.

- The paper primarily focuses on empirical results. That would be great if sufficient theoretical support could be provided for the observed improvements.(but I know this may be out of the scope of this paper.)

---

> ### Author Response · Authors · 2024-07-13
> **Respobse to Reviewer sdai**
>
> We thank the reviewer for the careful assessment of our work and the detailed feedback.
>
> In the revised manuscript, we emphasize the necessity of accurate uncertainty measures for achieving better results. Furthermore, we perform additional experiments using conformal prediction, which is widely used as a reliable uncertainty measure.  The results in Figure 7 of the revised manuscript also support our hypothesis under this different uncertainty measure. This demonstrates that there are uncertainty measures that can be integrated into our framework while achieving good performance. Moreover, uncertainty measurement is an important and growing research field, and better uncertainty measures will strengthen the performance of our framework.
>
> We have also outlined the future provision of theoretical analysis to support the empirical results of the paper.

---

> > ### Comment · Reviewer_sdai · 2024-07-25
> >
> > Thanks for the authors' response. I am satisfied and do not have any other questions.

---

### Decision · Action_Editor_5sBB · 2024-08-18

**Recommendation:** Accept as is

**Comment:**

This paper introduces FairDSR, a framework to improve the fairness-accuracy tradeoff in machine learning models when demographic information is incomplete or unavailable. The paper proposes to train an attribute classifier to predict missing demographic information and quantify the uncertainty of the classifier. The fairness constraints are only enforced on data with low uncertainty, instead of all data. Experiments demonstrate the advantages of the proposed method over multiple baselines.

The reviewers generally agree that the paper conducts solid experiments to demonstrate the effectiveness of the proposed method (all reviewers). Reviewers 6s5W and JCP5 further agree that the paper is well-written, easy to follow, and results are presented clearly. During the rebuttal, the authors addressed the reviewers' concerns regarding the dependence on reliable uncertainty measures, missing details of the method, and incomplete discussions with baselines. The reviewers acknowledged that their concerns have been addressed and recommended acceptance for the paper.

**Audience:**

Yes.

**Claims And Evidence:**

Yes.